# Natural Resource Conservation Based on Community Economic Empowerment: Perspectives on Watershed Management and Slum Settlements in Makassar City, South Sulawesi, Indonesia

**Batara Surya [1],\*** , **Syafri Syafri [2]**, **Hernita Sahban [3]** and **Harry Hardian Sakti [4]**

1    Departement of Urban and Regional Planning, Faculty of Engineering, Bosowa University, Makassar 90231, Indonesia
2    Departement Urban Ecology, Bosowa University, Makassar 90231, Indonesia; syafri@universitasbosowa.ac.id
3    Departement of Economic, STIM Lasharan Jaya, Makassar 90231, Indonesia; hernitasahban@stim-lasharan.ac.id
4    Departement of Urban and Regional Planning, Faculty of Engineering, University Muhammadiyah, Bulukumba 92511, Indonesia; revplano07@gmail.com
*    Correspondence: bataraciptaperdana@yahoo.co.id

**Abstract:** The purpose of this study is to analyze the influence of slum development, community poverty, and community behavior on environmental degradation in the Tallo river basin in Makassar City and to analyze the effects of natural resource conservation, economic empowerment, community capacity building on the productivity of economic enterprises and ecosystem-based sustainability. This study uses a qualitative-quantitative approach in sequence. Data were obtained through observation, surveys, and documentation. The research findings show that slums, poverty, and community behavior have a significant effect on the decline in the environmental quality of the Tallo river basin in Makassar City, with a coefficient of determination of 32.2%. The results showed that the conservation of natural resources, economic empowerment, and community capacity building were positively correlated to increasing the productivity of community economic enterprises and the sustainability of watershed ecosystems. The assertion is that watershed conservation, integrated with economic empowerment, contributes positively to economic, social, and environmental sustainability. This study offers the concept of conservation of natural resources based on community economic empowerment as a solution to the handling of slums for the case of metropolitan cities in Indonesia, to support metropolitan city development programs nationally.

**Keywords:** natural resource conservation; slums; economic empowerment; economic growth; sustainable development

## 1. Introduction

Urbanization and globalization are strategic issues affecting the dynamics of big cities and metropolitan areas today. Urbanization is a very complex phenomenon and its scope is very broad, covering social, economic, political, and geographic aspects. Urbanization is a global problem with demographic trends [1]. Urbanization includes many different factors, and many types of urban indexes can be used to indicate the process of urbanization. The urban population in the world will grow at a rate higher than the total world population [2]. By 2018, 4.2 billion people, around 55% of the world's total population, had become urban residents. This figure when compared with the 1950 period, the proportion will increase by 25%. It is estimated that around 64 percent of developing

countries and 86% of developed countries will experience urbanization. This figure is equivalent to around 3 billion city residents in 2050, which will be dominant in Africa and Asia.

In Western Europe, America, Australia, Japan, and the Middle East, as high-income countries, it is estimated that around 80 percent of the population will live in urban areas. Then, Eastern Europe, East Asia, North and South Africa, and South America, as middle and upper income countries, are predicted to be between 50% and 80%. In the UN report, it also mentions that around 2.5 million people will live in cities in the next 30 years and around 90 percent are in Asian and African countries. Even though the level of urbanization is quite low, the population of cities in Asia will reach 54%, while Europe and Africa, each around 13%. The explosion of the urban population will be concentrated in certain countries, namely 35% in India, China, and Nigeria, until 2050. That is, there will be an additional 416 million people for cities in India, 255 million in China and 189 million in Nigeria [3]. This phenomenon illustrates that industrialization and economic growth in large cities and metropolitan areas are the driving force and pull factor for the migration of rural populations into cities on a large scale.

Urbanization in Indonesia is predominantly triggered by economic development oriented to the industrial, trade, and service sectors. The sector tends to be located in big cities and metropolitan areas. Urbanization, as a process for the case of big cities and metropolitan areas in Indonesia, is very different from the case in developed countries. The combination of typical urban and rural activities has resulted in certain features of rural–urban transition as the urban population has continued to increase notably [4]. Industrialization and the service sector which continues to increase basically will require the support of the availability of utilities, the availability of water, electricity, ports, and airports that require skilled labor and markets. Furthermore, urbanization and economic growth in big cities and metropolitan areas in Indonesia are indicated to be triggered by the role of investment, both domestic and foreign.

Population migration is closely related to the desire of migrants to get decent work and improve family welfare. Potential gains in absolute income through migration are likely to play an important role in households' migration decisions, but international migration by household members who hold promise for success as labor migrants can also be an effective strategy to improve a household's income position relative to others in the household's reference group [5,6]. The new economics of migration supposes the view that population migration is not only influenced by the labor market, but there are other supporting factors, namely natural resources and ease of finding employment. These individual spatial mobility processes affect the future development of regional disparities [7]. One of the most important determinants of youth unemployment is the rate of economic growth [8,9]. Furthermore, urbanization is defined as "the demographic process whereby an increasing share of the national population lives within urban settlements [10,11].

The dynamics of the development of Makassar City as a core city in the Metropolitan Maminasata urban system show the same symptoms for the case of metropolitan cities in Indonesia, in relation to rural migration to urban areas. This condition was marked by an increase in the population of 1,429,242 people in 2014 and then increased by 1,508,154 people in 2019 with a growth rate of 1.28%. Increasing population and economic growth have an impact on the conversion of productive agricultural land and changes in the use of very complex spaces [12]. There are many factors contributing to this process, i.e., large-scale housing and new towns, industrial estates, and toll road development [13].

The position of Makassar City as a core city in the Mamminasata Metropolitan urban system and the increase in population due to urbanization and migration has an impact on spatial expansion towards meeting the need for large scale housing and settlements [14]. The Tamalanrea and Tallo Districts of Makassar City, as the object of study, are new areas that have been developed to meet the needs of the population in terms of preparing large scale settlements, shopping centers, education, infrastructure and other socio-economic activities. The growing complexity of spatial use has led to a decline in environmental quality and slum development in the Tallo watershed.

Slums that develop along the Tallo River watershed are synonymous with community poverty. Developing poverty and slum areas are assessed based on indicators: (i) limited access to land for

housing needs, (ii) difficulties in getting decent work due to inadequate formal educational background, (iii) limited access to basic urban services, (iv) un-patterned residential environments and potential fire threats high, (v) social problems and urban crime are quite high, (vi) environmental sanitation is very bad, and (vii) settlement quality is reduced due to the burden of environmental pollution.

Watershed conservation in the study area is an effort to optimize the use of natural resources to ensure their wise and sustainable use of supplies and to improve the quality of their diversity. Conservation of natural resources is oriented to the sustainability of the environment to ensure its use for the current generation and maintain its potential to ensure the needs of future generations' lives [15]. The conservation of natural resources in the Tallo watershed located in the Districts of Tamalanrea and the District of Tallo in Makassar City is focused on optimizing the utilization and sustainability of the utilization of natural resource potential based on community economic empowerment towards the sustainability of ecosystem management. Action for empowerment has been increasing within communitie saround issues of access to resources and entitlements, capacity building, the nurturing of leadership and local initiative and institutional development [16].

The conservation of natural resources based on community economic empowerment, is intended to encourage increased productivity of community economic enterprises, control environmental pollution, and maintain the sustainability of environmental ecosystems. Thus, this study is intended to answer research statements, (i) what is the influence of the development of slums, community poverty, and community behavior on the degradation of the environmental quality of the Tallo river basin in Makassar City, and (ii) is there a relationship/correlation between the conservation of natural resources, economic empowerment, strengthening community capacity towards increasing the productivity of economic businesses and community-based ecosystem sustainability.

## 2. Conceptual Framework

Excessive urbanization and maximum compaction in the dynamics of the development of Makassar City have an impact on the use of very complex spaces. Urbanization has an impact on reducing income inequality [17]. The complexity of spatial use and the inability of the community to access formal land and employment, has caused slums to develop in the Tallo and Tamalanrea districts that are dominantly located in the Tallo River watershed. Slum relocation and clearance were the conventional solutions based on a primarily negative outlook of slum settlements and a strong commitment to a modernist approach of building high-rise complexes to replace them [18]. The existence of slums in the study area has an influence on the deterioration of environmental quality and environmental pollution in watersheds, so the handling of natural resources based on community economic empowerment is needed. Research conceptual framework is shown in the following Figure 1.

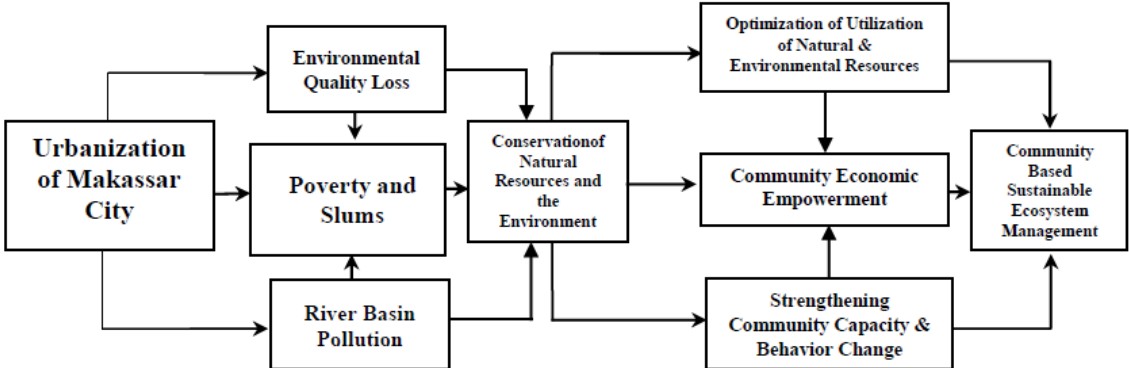

**Figure 1.** Conceptual Framework Natural Resource Conservation. Based on Community Economic Empowerment. Source: author elaboration.

### 2.1. Slums and Natural Resource Conservation

Settlements are environments that are equipped with workplaces, infrastructure and facilities, to support livelihoods and livelihoods, so that their functions can be effective, effective and sustainable. Environmental sustainability depends upon a balance between maintaining the capacity of ecosystems and satisfying the needs of society [19]. Slums that develop in the study area are characterized by a high population density and are dominantly inhabited by the urban poor.

The focus of the study of slums includes three main aspects, namely (i) physical condition based on the condition of residential buildings that are very tight with very low quality of construction, (ii) road network is not patterned and not hardened, (iii) public sanitation and drainage are not functioning, and (iii) sanitation and waste management have not been handled well. Furthermore, the social, economic, and cultural conditions of the communities living in the location are assessed based on: (i) the level of income is quite low, (ii) loose social norms, (iii) a culture of poverty that colors people's lives as seen from attitudes and behaviors that tend to be apathetic. These conditions have a relationship with poor health conditions, sources of pollution, sources of spread of disease, and deviant behavior, as well as its impact on people's lives as a whole. In particular, urban environments create highly heterogeneous socio-economic and environmental conditions that can affect the transmission of vector-borne diseases dependent on human water storage and wastewater management [20].

Conservation of natural resources in the study area is oriented to the preservation of the Tallo watershed and the balance of the environmental ecosystem, in order to support efforts to improve the welfare and quality of life of the local community. Biodiversity is essential for the functions and services of ecosystems on which humans depend, and is directly linked to the economic, social, and environmental components of sustainability [21]. Thus, the relationship between the existence of slums has a close relation to the conservation of natural resources and the preservation of the Tallo watershed ecosystem. That is, that the conservation of natural resources carried out is oriented to the role of community participation and community economic empowerment in a sustainable manner.

### 2.2. Economic Empowerment and Sustainability of Ecosystem Management

Community economic empowerment is contextualized in developing productive and competitive economic enterprises. Community empowerment is an effort to change a condition or condition of a community with a very low standard of living to a better condition in the economic, socio-cultural and political sense [22]. In the context of the area of study, economic empowerment is carried out through efforts to develop productive economic enterprises for the purpose of improving welfare while reducing poverty and unemployment.

The sustainability of ecosystem management is understood to be a socio-ecological process towards achieving the same ideals between the actors of development and the local community. This means that healthy ecosystems and environments are needed to support the continuity of society towards the optimization of sustainable management of resources and ecosystem protection. Social-ecological connections are not static, but change dynamically over time, and the consequences of this change in connection are often not linear and uncertain [23]. Furthermore, the sustainability of ecosystems in the study area is interpreted as an effort towards a balance in achieving environmental sustainability, related to climate change, environmental degradation, over consumption, economic growth and socio-cultural society. The increase in demand for ecosystem services is largely driven by population increases and changes in consumption patterns [24]. Thus, economic empowerment and sustainable management of the system in the study area will encourage increased productive economic efforts based on the management of Tallo watershed ecosystems towards the benefits of mutual and mutually beneficial human and environmental relations towards sustainable development.

## 3. Profile of the Study Area

The expansion of spatial and economic growth in Makassar City is a determinant factor that causes poverty and developing slums (see Figure 2). The selection of study sites in the Districts of Tamalanrea and District of Tallo, is based on: (i) the reality that develops in the field is not singular but plural, (ii) the inability of the community to access economic resources has an impact on poverty and slums developing along river basins, (iii) the potential and social capital of the community are very supportive for the implementation of natural resource conservation and economic empowerment, and (iv) the development of slums along the watershed has the potential to be a threat to flooding and increased environmental pollution in the watershed. This research was carried out over three time period periods (2010, 2015, and 2019). The aim is to compare the level of environmental pollution due to the complexity of spatial use and community behavior in relation to the existence of slum locations.

The geographical location of the study area of Tamalarea District is the coordinates of 119°29′21,567″ E 5°6′33,884″S, while the Tallo District area is located at the coordinates of 119°26′5,786″ E 5°6′56,696″S. Furthermore, the total population in the Tamalanrea District area in 2019 was 114,672 people covering 8 urban villages and slum areas that developed along the watershed area of 17.17 hectares. Furthermore, the population in the Tallo District area in 2019 amounted to 140,023 people, including 15 urban villages and slum areas that developed along the river basin area of 47.06 hectares. Population growth rates in the two regions are higher than the population growth in Makassar City, which is 5.83 percent. The dominant community work orientation, among others, includes: fishermen, odd jobs, online taxi, river transportation services, and pond farmers.

Slums along the Tallo watershed are developing due to rapid changes in spatial function and the inability of the poor to access high-value land and an inability to access the city's economic resources. Urban informal settlements, usually defined by certain criteria such as self-build housing, sub-standard services, and residents' low incomes, are often seen as problematic, due to associations with poverty, irregularity and marginalization [25]. The accelerated development of Makassar City has led to the conversion of productive agricultural land and land conversion in the Tamalanrea and Tallo Districts. This condition is marked by the allocation of space for commercial functions covering an area of 84.47 hectares, large scale settlements 249.72 hectares, services 18.96 hectares, educational facilities covering an area of 22.80 hectares, and health facilities covering an area of 6.58 hectares. The research location is shown in the following figure.

Slums that develop in the study area (see Figure 2 and Table 1) cover four urban villages in Tamalenrea sub-district and 12 urban villages in Tallo sub-district area. From a total of 16 kelurahan in the two sub-districts, the slums that develop in the Tamalanrea sub-district occupy 32.12 hectares or 1.0 percent of the 3,184 hectares. Furthermore, developing slums in the Tallo District area occupy an area of 108.69 hectares or 18.64 percent of the district area of 583 hectares. The results of observations made illustrate that the slums in the Tamalanrea District area are located in the Tamalerea Jaya Village, occupying an area of 8.40 hectares, Parangloe Village occupies a land area of 7.40 hectares, Bira Village occupies an area of 22.98 hectares, and Tamalanrea Village occupies an area of 7.43 hectares. Furthermore, slums in Tallo Subdistrict, located in Tallo Village occupying 30 hectares of land, Rappokalling Vilage occupying 11.18 hectares, Bunga Eja Beru Village occupying 11.62 hectares, Buloa Village occupying 1.26 hectares, Kaluku Bodoa Vilage occupies a land area of 19.21 hectares, Lakkang Village occupies an area of 11.57 hectares, Lembo Village occupies 4.53 hectares, Pannampu Village occupies an area of 18.16 hectares, Rappojawa Village occupies an area of 2.76 hectares, Suanga Village occupies an area of 1.04 hectares, Tammua Village occupies an area of 2.08 hectares, and Wala-Walaya Village occupies an area of 4.68 hectares.

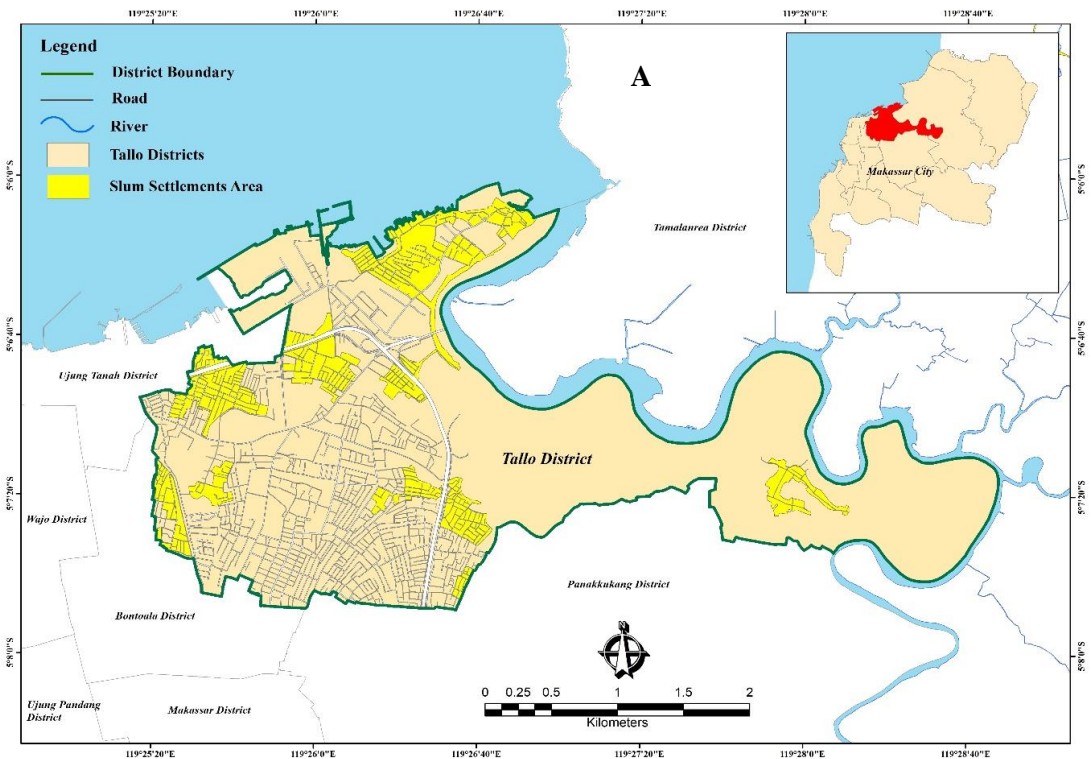

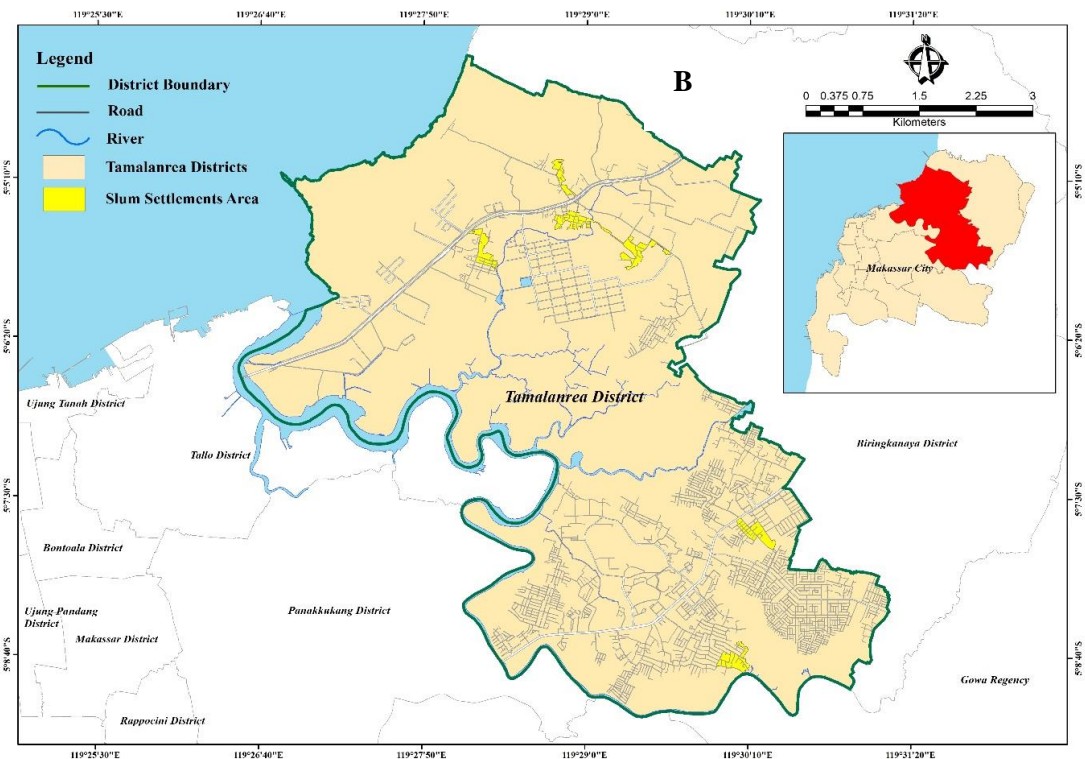

**Figure 2.** *Cont.*

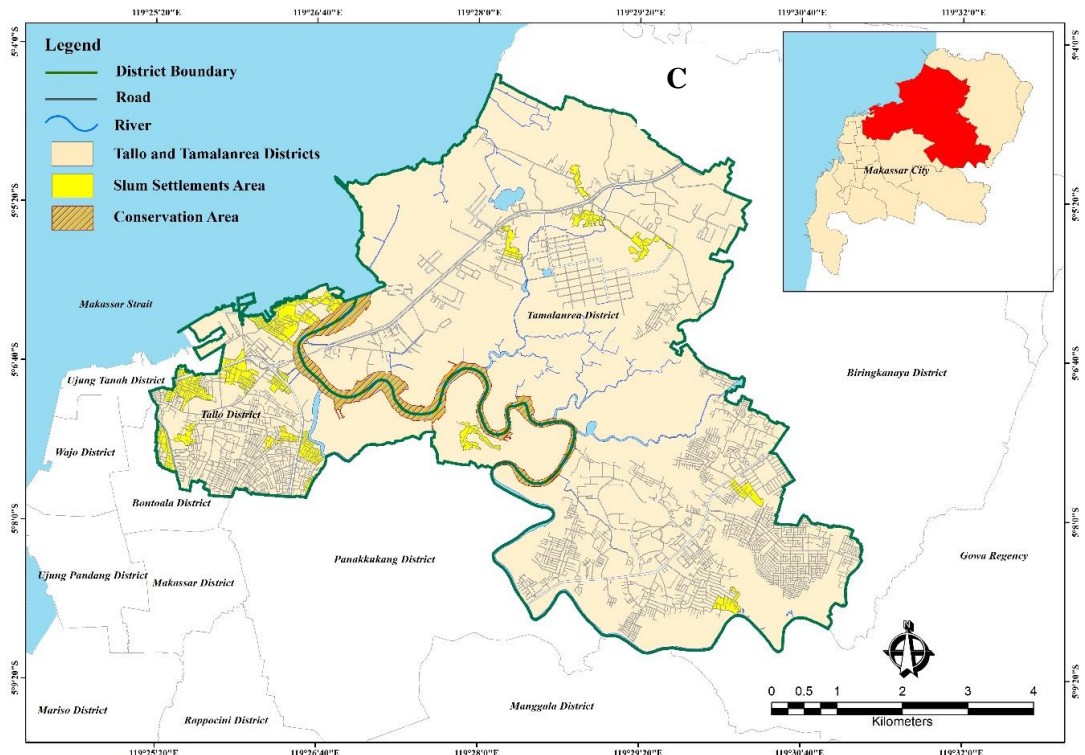

**Figure 2.** (**A**–**C**). Research Locations in Tamalanrea District and Tallo District in Makassar City.

**Table 1.** Geographical Location and Typology of Slums Based on Kelurahan in Tamalanrea District and Tallo District in Makassar City.

| Slums in Tamalanrea District | Coordinate Point | Slum Settlement Area | Typology of Slums |
|---|---|---|---|
| Tamalanrea Jaya | 119°30′3.489"E 5°8′40.309"S | 8.40 | Lowland |
| Parang Loe | 119°28′14.042"E 5°5′44.106"S | 4.68 | Lowland |
| Bira | 119°29′20.791"E 5°5′40.706"S | 16 | Lowland |
| Tamalanrea | 119°30′10.479"E 5°7′44.025"S | 3.04 | Lowland |
| **Slums in Tallo District** | | | |
| Tallo | 119°26′24.975"E 5°6′23.269"S | 27.55 | Lowlands, Water Edge, and above Water |
| Rappokalling | 119°26′37.093"E 5°7′23.848"S | 11.18 | Lowlands, Water Edge, and above Water |
| Bunga Eja Beru | 119°25′24.739"E 5°7′23.783"S | 12.16 | Lowlands and Water Edge |
| Buloa | 119°26′21.936"E 5°6′28.049"S | 1.26 | Lowlands, Water Edge, and above Water |
| Kaluku Bodoa | 119°25′58.907"E 5°6′46.529"S | 17.07 | Lowland |
| Lakkang | 119°27′53.154"E 5°7′16.404"S | 11.57 | Lowlands and Water Edge |
| Lembo | 119°25′33.218"E 5°7′20.345"S | 3.59 | Lowland |
| Pannampu | 119°25′35.505"E 5°6′56.154"S | 17.76 | Lowland |
| Suwangga | 119°25′38.288"E 5°7′13.514"S | 1.34 | Lowland |

**Table 1.** *Cont.*

| | | | |
|---|---|---|---|
| Tammua | 119°26′36.485″E 5°7′42.583″S | 0.8 | Lowland |
| Wala-Walaya | 119°26′23.819″E 5°7′16.747″S | 0.54 | Lowland |
| Rappojawa | 119°26′17.038″E 5°7′20.178″S | 3.87 | Lowland |

Sumber: Sorce Elabotaor, BPS Kota Makassar, Map[(c)] 2019 Google.

### 3.1. Data Colection Methods

This research is Naturalistic, Rationalistic, Holistic, Cultural, and Phenomenological [26,27]. Thus, the research approach used is sequential explanatory, which is a combination of quantitative and qualitative approaches in sequence. The qualitative method used to build hypotheses was based on phenomena that develop in the field. Meanwhile quantitative methods are descriptive, comparative, and associative and test hypotheses based on the results of qualitative studies.

Data collection techniques used in this study: observation, in-depth interviews, questionnaires, and documentation. In-depth interviews were used to gather information related to the development of slums, community poverty, and community behavior towards the degradation of the environmental quality of the Tallo watershed. This information, supported by observations made on several things, namely (i) the characteristics and typologies of slums, (ii) pollution of the Tallo watershed, (iii) the potential for community economic enterprises that can be developed and (iv) conditions of community poverty, and (v) community behavior towards environmental conditions.

In accordance with the focus of the study, the data sources in this study are: (i) to obtain data on the influence of the development of slums, community poverty, community behavior towards environmental degradation in the Tallo watershed, the data source is the intensity of land use, spatial patterns, facilities and infrastructure as well as the development of slum areas in Tamalanrea and Tallo Districts. The data is obtained through field observations and documentation, (ii) To obtain data on natural resource conservation, economic empowerment, community capacity building towards increasing the productivity of economic enterprises and community-based ecosystem sustainability, the data source is a conservation program conducted by the Makassar City Government obtained through documents, surveys and in-depth interviews. From this data, a descriptive situation, event, and observable behavior were carried out in detail.

### 3.2. Selection of the Respondents

The sample is determined by a purposive method. In this case, the researcher determines the sample based on the special characteristics of the population, in accordance with the research objectives to be achieved. The determination of the sample is based on considerations: (i) the population has been determined from the beginning of the study, (ii) the sample was determined based on the main characteristics of the population, and (iii) the sample determined in this study included 350 family heads of communities located in the slums of Tamalanrea and Tallo Districts.

The method of selecting respondents was followed directly by researchers at the location of slums in the Tamalarea and Tallo Districts. The aim is to understand the situation and condition of the object of research. The determination of key informants in this study was used to collect qualitative data. The determination of the informant was done in a snowball manner, meaning that the researcher determined the people who could be interviewed based on information provided by the population, who could provide good information about the existence of slums at the study site. As an informant the researcher established one of the community leaders. Furthermore, information from the informant is then forwarded to other community leaders who can be interviewed, until the information provided has gotten the same picture. Thus, the role of the informant is used to explore some of the questions that will be answered in a questionnaire that requires a more detailed explanation. This informant is called by the researcher as the perpetrator in the phenomenon under investigation.

Quantitative data were collected from respondents or research samples. The sample is determined by a purposive sampling technique on residents located in slums which are by researchers according to specific characteristics. The characteristics referred to by the researcher are that the sample must be a resident who is in a slum location, has a family, lives or has not moved for at least five years, and knows the conditions of the local area. The characteristics stated are used by researchers in determining respondents. The results of filling out the questionnaire were analyzed using non-parametric statistics, especially the percentage analysis based on frequency numbers. Sampling refers to [28].

$$n = \frac{n}{Nd^2 + 1} \tag{1}$$

Information:

n = Sample

N = Population

d = Error rate (0.5) or 5% of the 95% confidence level.

Each village has a different number of respondents determined by certain considerations. That is, it really depends on the number of populations that are in each slum location. Determination of the number of samples is carried out proportionally based on work orientation, educational background, and economic efforts developed by the community. Thus, the general reason for determining the number of respondents in each kelurahan is the number of family heads located in the slums of Tamalanrea and Tallo Districts.

### 3.3. Observation, Survey, and Focus Group Discussion

Observations made in this study are associated with two things, namely information and context. Information collected through observation in accordance with questions raised in interviews and in-depth questionnaires, such as watershed coverage before slums develop, changes in land use functions, where people live in group ties, patterns of social relations that occur, institutional systems, stratification social, social status and class and community values that can be used in the conservation of the Tallo watershed. Data collection strategies carried out in the field are divided into two categories, namely (i) entering the field, in this process intending to explore and understand the situation, and study the circumstances and background of the people who are subject for the purpose of improving the relationship of the researcher with the subject under study, so that it runs harmoniously, (ii) participates while collecting data. This process is carried out for the purpose of finding concepts that will be elaborated in accordance with research objectives.

The survey in this study, using a questionnaire instrument. The questions formulated in the questionnaire were based on the results of preliminary interviews with several community leaders. Then added to the observations and documentation obtained previously and the concepts related to the research objectives. The questionnaire in this study was used for two functions: (i) descriptive, and (ii) measurement. The purpose of using a questionnaire is to provide a description of some characteristics of individuals, or groups of people. The questionnaire was used to collect data, among others: level of education, social stratification, type of work, economic effort, level of income, patterns of social relations within community groups, community institutions, and social problems that occurred, related to the conservation of the Tallo watershed.

Questionnaires were distributed to 16 slum locations within the Tamalanrea and Tallo Districts. The reason the researchers set the location is based on the development conditions of the two sub-districts are quite significant and the transfer of land use is quite intensive. Criteria for the actors who filled out the questionnaire (respondents) were residents who were married, had at least lived permanently or did not leave the place for a period of five years. Frequency values of the two concepts above were collected through a survey using a questionnaire and carried out by an enumerator. Furthermore, the data collected by the enumerator is then processed using a frequency table and using

percentage analysis. The results are discussed qualitatively through interpretation of meanings to expressions, which in the end this study can be deeper.

Focus group discussions are held to provide convenience and opportunities for researchers to establish openness, trust, and understand perceptions, attitudes, and experiences of respondents. In this case the FGD was carried out through discussions that were carried out systematically and directedly related to the community's participation in the conservation of the Tallo watershed. Defining focus group discussin is a systematic process of collecting data and information about a specific problem that is very specific through group discussion [29]. FGD in this study was carried out 4 times, 1 time was carried out in the District of Tamalanrea and three times was carried out in the District of Tallo. The focus of the FDG is more directed towards increasing the role of the community in the conservation of the Tallo watershed and the possibility of economic efforts that can be developed.

### 3.4. Secondary Data

Documentation will be used throughout the study, including documents on land ownership structure obtained from the Makassar City Regional Development Planning Agency, data on population numbers obtained from the Makassar City Statistics Office, community poverty profiles obtained from the District Offices. All of these documents will be used to support in-depth interview data and observation and the creation of a research questionnaire. Secondary data collected by researchers for the purpose of completing the analysis include: Makassar Municipality in Figures 2019, and documents on the conservation concept of the Tallo watershed from the Makassar City.

### 3.5. Data Analisis

Data analysis is done by combining qualitative and quantitative data analysis. That is, the steps used for qualitative research at the same time are also used quantitative research. At the time of interpretation or analysis, each data is carried out reduction, namely for qualitative data categorization and quantitative data, descriptive statistical calculations are performed. The two data are then performed triangulation or between methods. This means that data obtained using a questionnaire will be explored deeper through two methods, namely qualitative and quantitative. This merger is more to strengthen the validity of the results of the analysis. Furthermore, the intended data reduction is grouping or categorizing data according to the scope of the study. Likewise, with the questionnaire, all questions made refer to the focus of the study. Regression analysis in this study was used to examine the effect of $X_1$ (slums), $X_2$ (community poverty), $X_3$ (community behavior) on Y (degradation of watershed environmental quality). The following equation was used:

$$Y = a + b_1X_1 + b_2X_2 + b_3X_3 + \ldots + b_nX_n \tag{2}$$

Furthermore, path analysis, refers to variables, namely: (i) $X_1$ exogenous independent variable (conservation of natural resources), (ii) $X_2$ exogenous independent variable (economic empowerment), (iii) $X_3$ exogenous independent variable (strengthening community capacity), (iv) Y endogenous dependent variable (productivity of economic enterprises), and (v) Z endogenous dependent variable (ecosystem sustainability). Path diagram using structural equation:

$$Y = PYX_1 + PYX_2 + PYX_3 + e_1. \tag{3}$$

Path analysis is used with consideration: (i) research metric data are measured based on interval scale, (ii) exogenous and endogenous dependent variables refer to multiple regression models, while intermediate variables refer to mediation models and the combined mediation and multiple regression models which are complex models, (iii) the relationship between variables is one-way, and (iv) cause and effect is based on the theory that there is a relationship or correlation between natural resource management, economic empowerment, community capacity building towards increasing the

productivity of economic enterprises, and the sustainability of community-based watershed ecosystems. The flow of the research process in the following Figure 3.

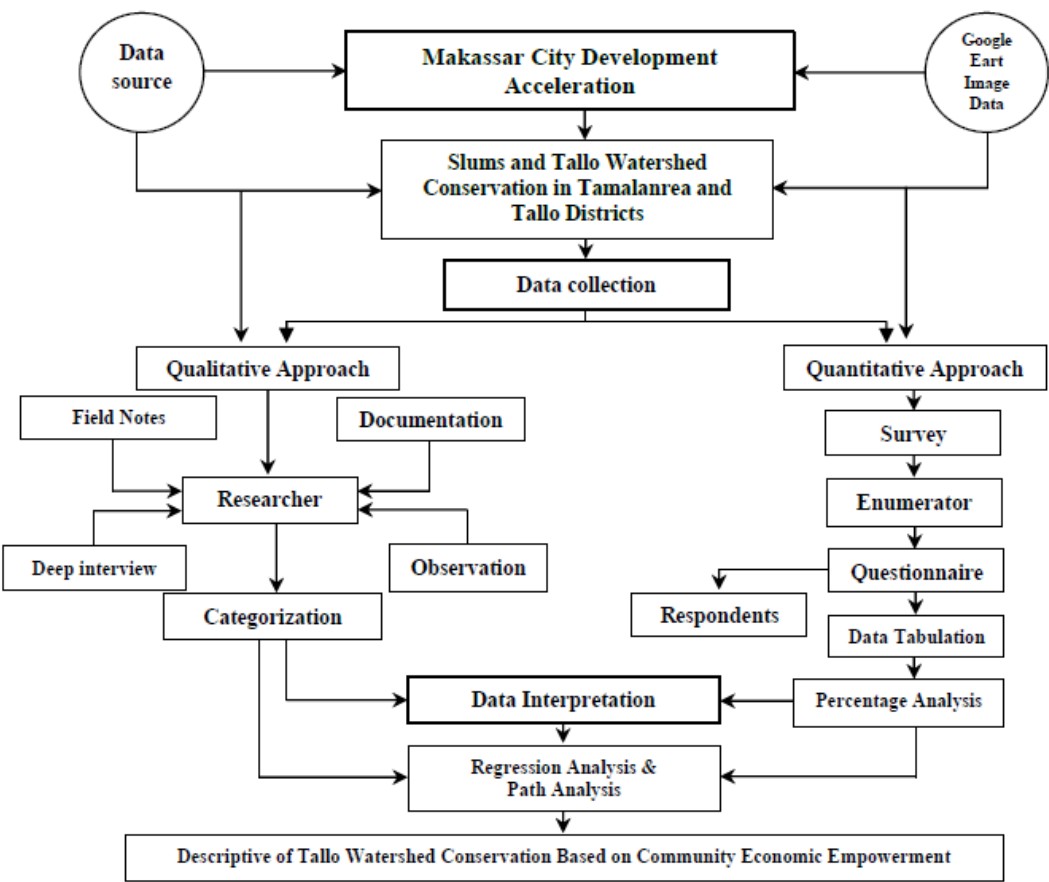

**Figure 3.** Research Process Flowchart, Natural Resource Conservation Based on Community Economic Empowerment. Source: author elaboration.

## 4. Result and Discussion

The population of Makassar City period (2015–2019) showed an increase of 78,912 people with a growth rate of 5.23 percent. This figure is higher with the national population growth rate. Over the last 20 years, many urban areas have experienced dramatic growth, as a result of rapid population growth as the world's economy has been transformed by a combination of rapid technological and political change [30]. Agglomeration economies and human capital are also important drivers of city growth [31,32]. Furthermore, the growth rate of the poor population in Makassar City in the 2015–2019 period was 1.03% [33]. This figure confirms that excessive urbanization and maximum compaction towards economic growth are positively associated with increasing poverty in Makassar City.

### 4.1. Poverty and Slums in Makassar City

Excessive urbanization and maximum compaction in line with an increase in community poverty is one of the factors causing the development of slums. The poverty rate in Makassar City is shown in the following Figure 4.

Figure 4 shows that the dominant poverty rate is located in the Tamalate District area of 5193 households or 34.8% of the total population in Makassar City. Furthermore, the lowest poverty rate is located in the UjungPandang District area of 216 households. This figure illustrates that the existence of poor people in Makassar runs parallel to the development of slums, urban crime, environmental degradation, and other social problems. The proposed interpretation is related to the causes of

poverty in Makassar City, which can be explained by the (i) inability to access land and access to production, (ii) inability to access business capital to develop economic businesses, (iii) pressure on poverty and disobedience of the community, in the sense that they are relatively isolated and do not have sufficient access to obtain information that is needed, and (iv) inability to access formal employment, due to low educational background and limited expertise. That fact confirms that the poverty rate, which tends to increase in Makassar City, causes slums to develop on illegal lands and specifically in watersheds. The main characteristics of slums are characterized by: poor environmental and infrastructure conditions; limited access to services due to lack of income to pay for care and prevention services; and relies on poor quality and most informal and unregulated health services that are not suitable to meet the unique realities and health needs of slum residents [34].

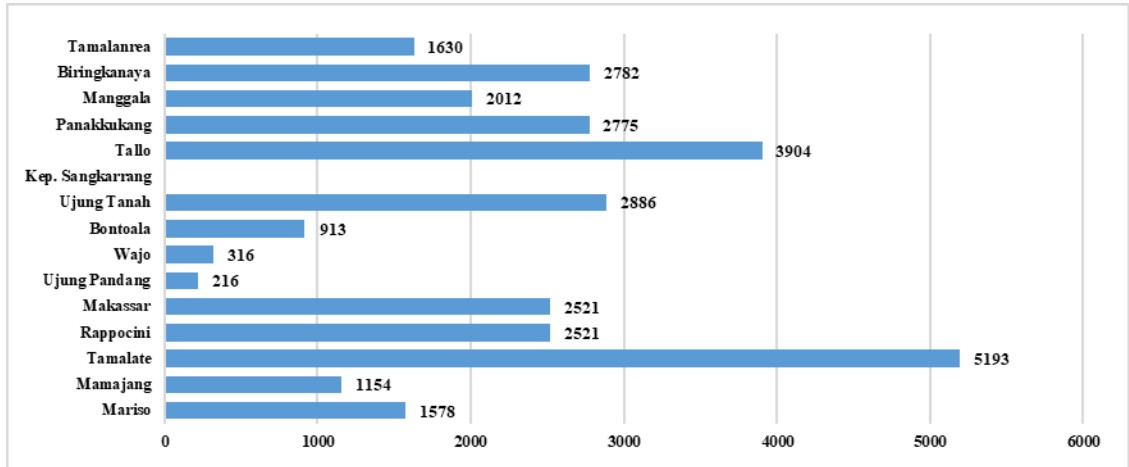

**Figure 4.** Number of Poor Residents in Makassar City in 2019.

Increased economic productivity and the expansion of urban sprawl in Makassar City that continues to increase towards the Tamalanrea and Tallo Districts are positively associated with the conversion of productive agricultural land and slums to develop along the Tallo watershed. The increasing increase in poverty has caused the inability of poor people to provide adequate and adequate housing facilities for their homes and the choice to build residential facilities located along the Tallo watershed. Field facts found indicate that poor urban settlements that develop, in addition to having an impact on reducing land cover, damage mangrove forest habitat, as well as watersheds, which are used as a sewerage system for households and latrines that are channeled directly into the river. The dominant river is utilized as a giant septic tank by most household and dispose of household waste and latrines directly into the river [35]. The distribution of Makassar City slums is shown in the following.

Figure 5 shows that 164.8 hectares, slum areas are predominantly located in the Tamalate Subdistrict area and are distributed in nine urban areas with the dominant slum categories being medium slum and light slum, while based on the typology of slum that develops, it is divided into three categories, namely settlement slums in the lowlands, slums on the water's edge, and slums on the water. Furthermore, the slums with the smallest area are located in Mamajang District, which are distributed in two villages with a total area of 5.97 hectares. The level of slums in the Mamajang Subdistrict is categorized as mild slum with lowland slum typology and above water. Increasing poverty rates that continue to increase are positively associated with slums that develop in 15 sub-districts and are distributed to 127 urban villages in Makassar City. This fact confirms that the weak control of spatial use and the dynamics of the rapid and revolutionary development of Makassar City in line with urban urbanization, as well as the poor access of the poor to formal employment causes slum settlements to increase from time to time. Interpretations that can be submitted related to these conditions, namely (i) urban development

policies that favor the poor have not been optimally implemented, and (ii) limited access to urban economic production for poor and marginal groups of people.

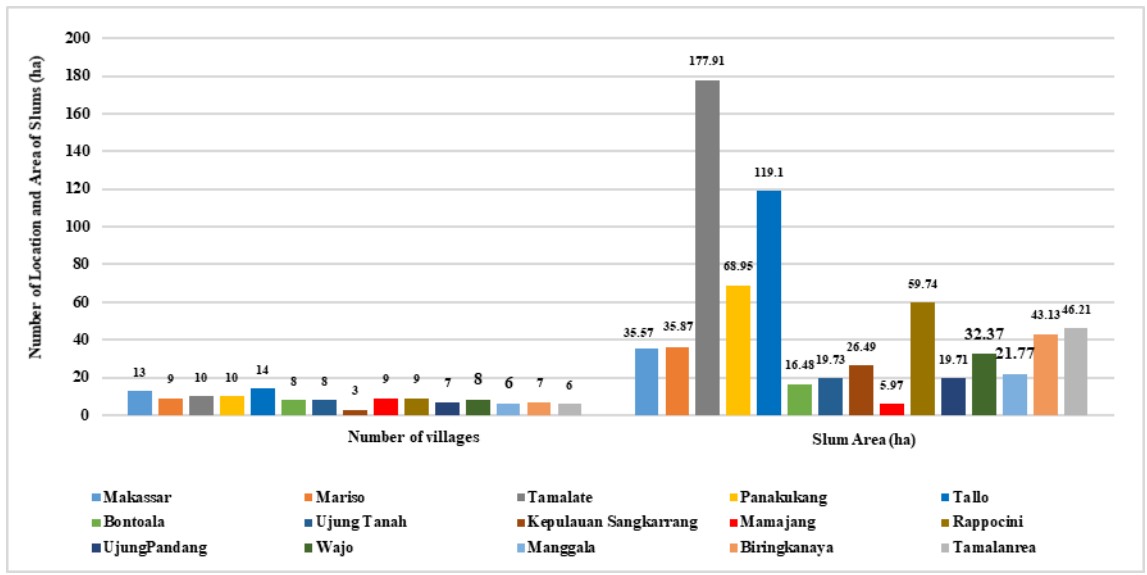

**Figure 5.** Distribution of Makassar City Slums, 2019.

The local government is unable to manage urbanization, and migrant workers without an affordable place to live in dwelling in slums [36,37]. Slum areas that develop in Makassar City, especially those located along river basins have a positive contribution to environmental degradation. The condition is assessed based on indicators; (i) water quality pollution due to household waste, (ii) the rate of erosion and sedimentation of the river is quite high due to the reduction of mangrove habitat, and (iii) urban flooding. This fact confirms that the slums that develop in line with the increased urban activity in Makassar City are positively associated with decreasing environmental quality and increasing pollution in the Tallo watershed.

Urban slum settlements are generally excluded from public-sector resources, severely limiting access of residents to formal education, healthcare services, and water and sanitation [38]. In addition, slums have high densities, more educated, and youthful populations that can amplify the impact of public health interventions [39]. These results confirm that urban expansion and weak development control due to an increase in population have an impact on poverty and developing slums on illegal lands. Informal settlements are what have emerged either illegally on government land or on private land carelessly [40,41].

*4.2. Characteristics and Typologies of Slum Settlements in the Study Area*

Slums that develop in the Tamalanrea District area consist of three categories, namely (i) land slums covering an area of 20.59 hectares, and (ii) slums on the waterfront covering an area of 17.17 hectares. Furthermore, in the District of Tallo, there are three categories, namely (i) land slums covering an area of 15.70 hectares, (ii) slums on the waterfront covering an area of 17.44 hectares, and (iii) slums on water covering an area of 11.85 hectares. The existence of slums is characterized by unhealthy environmental conditions, high environmental pollution and environmental facilities and infrastructure that do not meet service standards. Unhealthy living conditions are the result of a lack of basic services, with visible, open sewers, a lack of pathways, uncontrolled dumping of waste, polluted environments, etc. [42]. The characteristics of slums and the typology of slums in the object of study are shown in the following Figure 6.

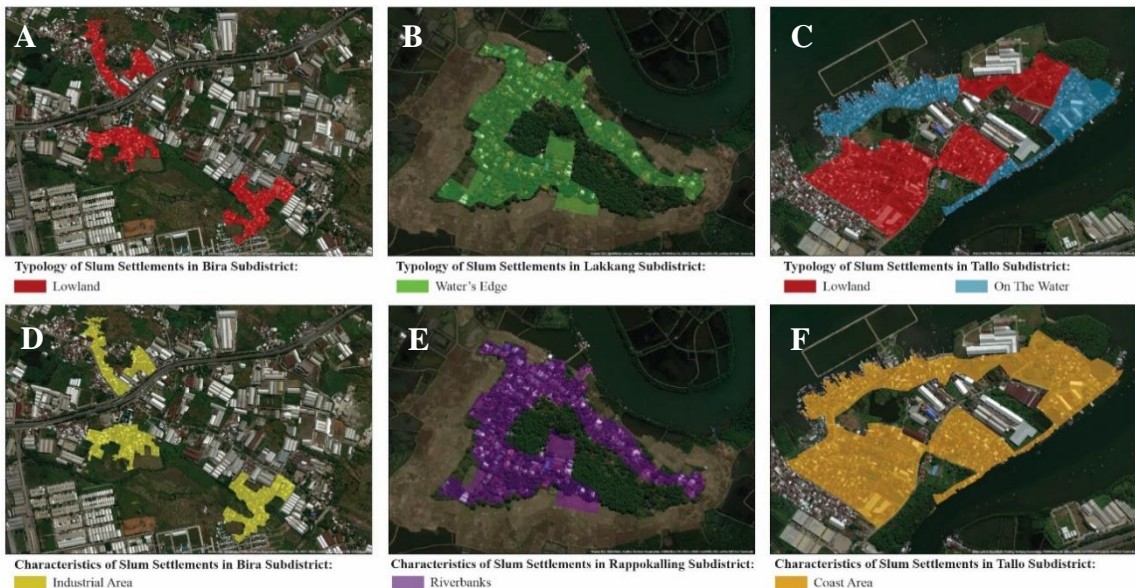

**Figure 6.** (**A–F**), Example, Characteristics and Typology of Slum Settlements in Tamalanrea District and Tallo District.

Figure 6 shows the slums that developed in the study area, namely (i) slums in the Tallo District area, spread over 12 villages. Furthermore, the slums that developed in the Tamalanrea District area are distributed in four villages. Slums in these two sub-districts are mostly located in watersheds and are in direct contact with the strategic functional areas of Makassar City, namely the industrial estate, trade center and education center. The existence of slums in Tallo and Tamalanrea sub-districts dominates illegal land, settlement infrastructure services, poor sanitation, and is very vulnerable to the threat of flooding. Characteristics and typologies of the slums that developed in Tamalanrea District and Tallo District. The characteristics of slums in the Tallo district are divided into two categories, namely (i) heavy slum areas of 27.55 hectares, and (ii) moderate slum areas of 81.14 hectares. Furthermore, the slums that develop in the District of Tamalanrea are categorized as moderate slums with an area of 32.12 hectares. Field facts found show that the existence of slums in these two sub-districts predominantly develops along the river flow. This means that in addition to the potential threat of urban flooding, the location has the potential to spread disease, including other social problems. What is interesting in this study is that the Tallo river by local people is part of their lives to make a living to meet the daily needs of their families. In addition, poor people living in slums in these two sub-districts have an average income of between $150–200 per month. This picture illustrates that the poor in the Tamalanrea and Tallo Makassar slums are identical with a low educational background, precarious work orientation, and an inability to access adequate housing.

In general, the characteristics of the typology of slums that develop in the two sub-districts are divided into four main categories: (a) Typology of slums that develop in low-lying locations. These slums are located in a radius of between 150–300 meters from the Tallo watershed and in direct contact with formal settlements built by the developer. The orientation of the main activities of the people who live in slums is dominant in the urban informal sector. (b) Slums are pulled over water. This means that the dominant slums that develop along the Tallo riverbank and occupy illegal land and have the potential to experience urban flooding every year, with poor environmental sanitation conditions and an inadequate drainage system for draining rainwater. The work orientation of the community at this location is predominantly day laborers, fishermen, and construction workers. (c) Slums that thrive on water. That is, slums that develop in the waters of the Tallo watershed. The dominant community work orientation at this location is fishermen and people who depend their lives on ferry and boat rental and boat transportation facilities. (d) Slums are developing around

industrial areas and trade centers. That is, slums are in direct contact with the strategic functional area of Makassar City. The dominant community work orientation includes, among others, industrial workers, odd jobs, garbage collectors, pedicab drivers and other non-formal urban activities.

Referring to the typology and characteristics of the slums that develop in the Districts of Tamalanrea and District of Tallo, the dynamics have a direct impact on environmental degradation and urban social problems. This means that the existence of slum settlements is faced with uncertainty in terms of location and land being utilized in relation to the Makassar City spatial plan, disaster prone, inadequate infrastructure conditions, social conflicts, and urban crime. Informal settlements and urban informality is a serious and common problem in third world countries and city density includes areas where environmental impacts occur [43,44].

Figure 7 show the population in the Tallo District area. Some things that can be explained, namely (i) Lakkang Village has the largest area or 115 hectares, the smallest area is located in Wala-Walaya Village or 11 hectares, (ii) the highest number of family heads is located in Kaluku Bodoa Village 4633 households, the lowest number of family heads is located in Lakkang Village or 244 households, (iii) the highest number of residents is located in Kaluku Bodoa Village with 22,753 inhabitants, while the lowest population is in Lakkang Village with 973 inhabitants, and (iv) the highest population density is located in Lembo Village with 897 inhabitants/hectares and density the lowest population is located in Lakkang Village eight people/hectares. Slums that develop in Tallo Subdistrict are based on their typology: lowland slums, waterfronts, and above water. The total population in the District of Tamalanrea. Some things that can be explained, namely: (i) the largest area is in Bira Village 9.26 hectares, the lowest is in Tamalanrea jaya Village 2.98 hectares, (ii) the highest number of family heads is in Tamalanrea Village 10,319 people, the lowest number of family heads is located in Parangloe Village 2396 inhabitants, (iii) the highest number of population is located in Tamalanrea Village 36,274 people, the lowest number of population is located in Parangloe Village 6.808 people. Furthermore, the typology of slums that develop are categorized as lowland slums.

The increasing population growth tends to increase the need for formal housing and developing slums in the Tallo and Tamalanrea districts. The health of the urban poor or slum dwellers is primarily due to crowding and lack of access to basic services, such as water and sanitation. Consequences of these living conditions include stress due to crowding, insecurity due to a lack of housing and land tenure, various types of illegal or criminal activities, including violence, drug-use, prostitution, etc. [45]. Increasing the number of residents running parallel with slums and the inability of the urban poor to get decent work and housing prices built by developers is high enough to cause people to break through vacant land and riverbanks and then use it to build residential facilities with improper building conditions habitable. In its development, this has an impact on the deterioration of the quality of the living environment and pollution of the Tallo watershed. Slums pose a significant challenge for urban planning and policy as they provide shelter to a third of urban residents [46].

The facts found in the field show that the existence of slums is very vulnerable to health problems, poor sanitation, high building density, urban crime, and environmental facilities and infrastructure that do not meet the requirements. The cause of such environmental degradation is the decreasing amount of land covering along the River Course Zone [47]. Furthermore, ideas are needed to deal with slums, capacity building, slum dialogue, community participation, and infrastructure development [48]. The interpretation proposed as a solution to the problem that has been explained, namely (i) prevention efforts are needed, including the act of supervision and control as well as technical standards for the feasibility of residential buildings, (ii) community empowerment, the implementation of which is through the process of mentoring and information services. Furthermore, efforts are needed to improve the quality of the environment by means of restoration in this case is improvement, rebuilding into habitable settlements. Environmental rejuvenation is carried out through better settlement environment arrangement in order to protect the safety and security of the community around the watershed.

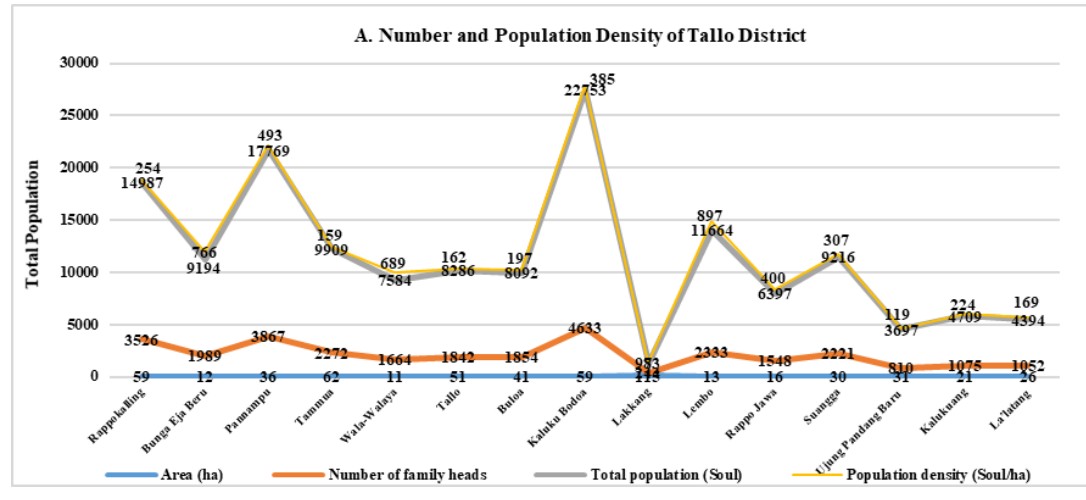

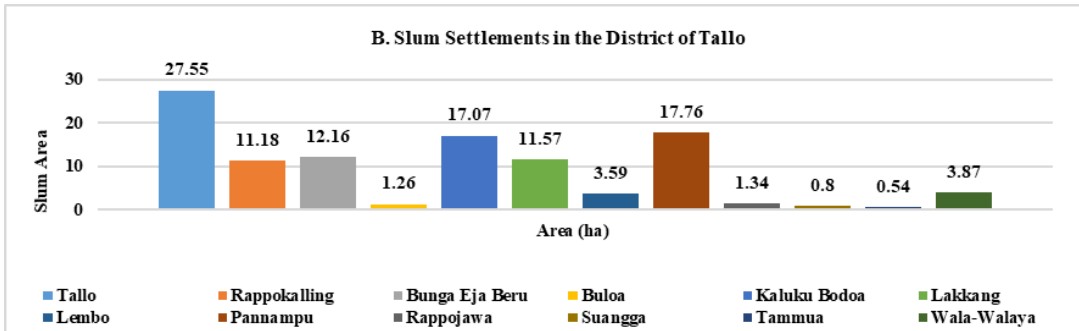

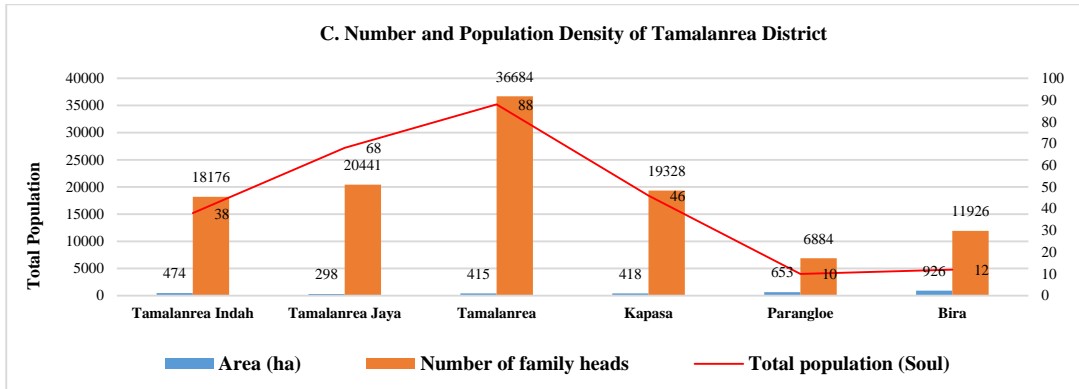

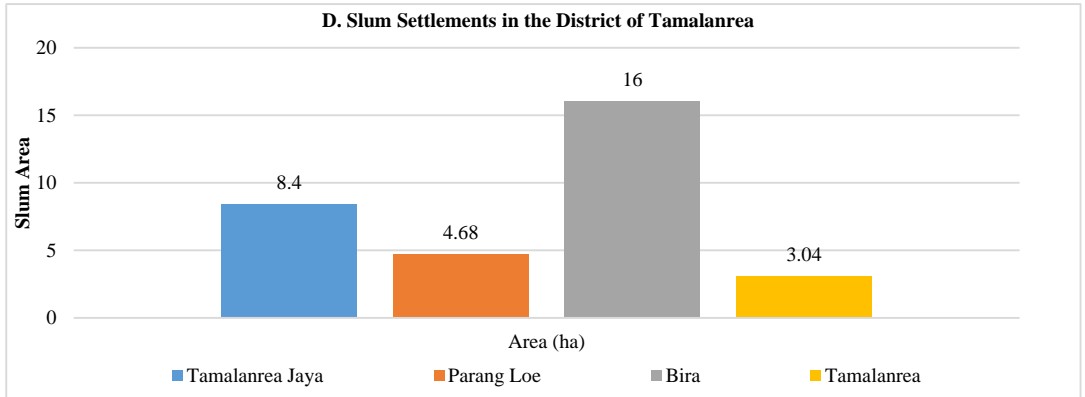

**Figure 7.** Number of Population and Area of Slums in Tallo District and Tamalanrea District. Source: Makassar Central Statistics Agency [33].

*4.3. Causes and Control of Environmental Pollution*

Increased urban activities and slums that develop along the Tallo watershed cause a decrease in environmental quality and increase water pollution, river silting, erosion and reduction of mangrove forest habitat. Erosion in advanced stages can not only transport sediments but also remove nutrients, organic carbon and agrochemical contaminants to outside the systems, resulting in a decrease of overall land productivity [49]. Furthermore, increasing pollution and decreasing environmental quality can be explained based on consumer habits and the level of industrialization between the community and the economy as well as due to inadequate waste management [50]. Conservation efforts are needed through controlling spatial use by limiting the development of slums along the Tallo watershed.

The function of the Tallo watershed is very strategic to support the development activities of Makassar City in the future, for the monitoring of water transportation facilities, clean water sources and urban flood control. The facts found in the field show that the source of water quality pollution in the Tallo watershed, namely, domestic waste originating from residential areas, offices, trade and other urban activities. Efforts to control water quality pollution are needed, namely: (i) water quality monitoring, (ii) determination of water pollution load capacity, (iii) determination of wastewater quality standards, (iv) preparation of wastewater treatment plants, and (v) inventory and identification of sources of water quality pollution.

Strategic steps that can be taken in the context of handling and controlling towards optimizing the utilization of the Tallo watershed, namely: (i) improving the quality of degraded land cover through efforts to rehabilitate mangrove forests based on land carrying capacity and land suitability classes, (ii) building awareness and changing people's behavior to use and maintain natural resources wisely, through counseling or training land use throughout the area Tallo river flow, (iii) construct and maintain man-made buildings along watersheds to control water flow to drain into waterways and rivers; and (iv) manage and change people's behavior not to throw garbage in the river, to avoid clogging of waterways and their impact on clean water supply.

These four strategic steps will require government policy support and community participation. Government policy support in the form of conservation-based regulations, control of spatial use, implementation and evaluation of development activities and encourage increased active community participation in supporting conservation and rehabilitation of mangrove forests, controlling environmental pollution, and monitoring water quality pollution both in quality and quantity [51].

Figure 8 shows the conservation of natural resources carried out in the study area. Conservation is oriented to the rehabilitation to restoration of the function of mangrove forests for the benefit of: (i) preventing coastal erosion and abrasion, in the sense that mangrove and mangrove trees have long stick roots so that they can protect the soil from erosion of tides; (ii) preventing sea water intrusion, in this case tree roots mangroves will be able to settle or hold mud so that it can prevent water intrusion. The conservation is carried out along the Tallo watershed towards the Makassar Strait coast by involving the role of the local community. The aim is to maintain the coastline and border lines of the Tallo river to be stable, protect the coast and rivers from the danger of erosion and abrasion, withstand storms or strong winds from the sea, and withstand the results of the mud accumulation process.

There are three categories of environmental pollution in the Tallo watershed within the Tamalanrea and Tallo Districts, namely: (i) surface water pollution due to household waste, industrial waste, rubbish, commercial activity and hospital waste, (ii) air pollution due to industrial pollution and vehicle fumes, and (iii) land pollution due to land reclamation and house waste stairs. One of the factors is the weak criminal law enforcement, especially in the environmental sector, so that in the implementation of many cases environmental pollution and/or destruction is not resolved completely [52].

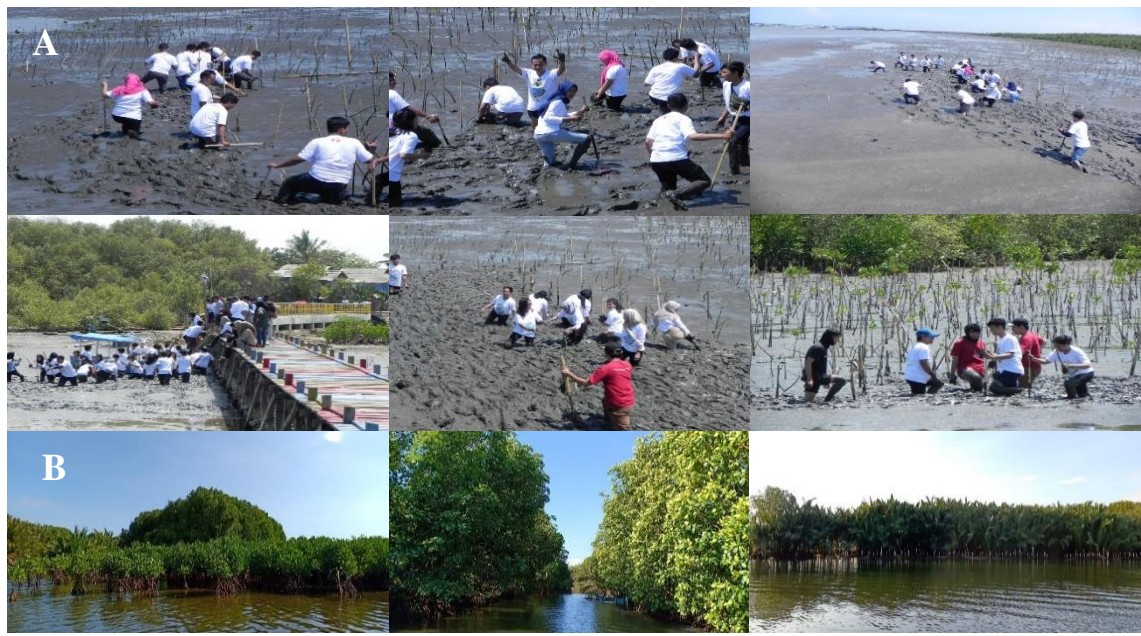

**Figure 8.** (**A**,**B**), Example, Rehabilitation of Mangrove Forests along the Tallo River Basin. Source: author elaboration, and Primary Data.

Field facts found confirm that the decline in environmental quality and the burden of environmental pollution due to increasing urban activity have an impact on decreasing river water quality, soil quality and air quality. Environmental problems, including outdoor and indoor air pollution, lack of water and pollution, desertification, and soil pollution, have become clearer and put the population at significant health risks [53]. This result, after being confirmed in the field, shows that the increase in environmental pollution in slums and its impact on the Tallo River watershed is affected by poor environmental sanitation conditions and waste management systems that have not been handled properly and damage to mangrove forest ecosystems. Furthermore, the results of field observations show that the Tallo watershed is currently used by the local community for bathing, washing and toilet needs, as well as being used as a means of crossing transportation which leads directly to the Makassar Strait. As a result, the existence of people who are on the riverbank is very susceptible to skin diseases and other diseases. Chemical pollution of surface water can create health risks, because such waterways are often used directly as drinking water sources or connected with shallow wells used for drinking water [54].

Conservation of natural resources carried out in the study area related to mangrove forest rehabilitation, divided into two categories. First, Figure 8A shows the situation of rehabilitating a mangrove forest that was damaged. The rehabilitation of mangrove forests that suffered damage is predominantly located on the Tallo riverbank and in direct contact with the Makassar strait coastline. Efforts to handle this are through reforestation and rejuvenation of mangrove forest habitats by involving the role of the community, non-governmental organizations, and the involvement of universities. This effort was carried out for the purpose of providing an understanding to the local community of the value of benefits obtained, among others; erosion control and sedimentation in watersheds and coastal areas in addition to the interests of saving the environment and integrated with tourism development through the support of the construction of a footpath along the rehabilitated Mangrove forest. Secondly, Figure 8B shows the the condition of mangrove forests that are still maintained. In this context, the efforts made through socialization to the local community to understand the benefits of mangrove forests that function. Furthermore, the handling steps taken are in the form of supervision and control followed by the role of community participation as part of efforts to save the environment.

Figure 9 shows the pollution of the Tallo watershed during the periods of 2010, 2015 and 2019. The dominant pollution potential is sourced from residential activities, high volume of waste generation, land reclamation, and waste disposal from urban activities. The indication is that the use of the Tallo river area has so far not taken into consideration the impacts and consequences. Pollution of the Tallo watershed which continues to increase based on its source is divided into three main categories, namely: (i) domestic waste originating from residential settlements, offices, trade and other urban activities; (ii) industrial pollution loads whose waste is directly channeled into rivers, (iii) ongoing land conversion due to settlement development, and pollution originating from water transportation, given the Tallo river by the local community is also used as a means of crossing transportation. If the condition is not improved, the carrying capacity of the environment will decrease and have an impact on river water quality going forward. Thus the management of the Tallo watershed is very important to be integrated with a variety of activities that have developed and management is carried out through the support of government policies, in this case the zoning of the river benefits consistently and the handling is carried out in an integrated and cross-sectoral manner.

The high burden of pollution and environmental damage will require the support of rescue and control of spatial use along the Tallo watershed. Urban land use was the key factor affecting water quality change, and limiting point-source waste discharge in urban areas during the dry season would be critical for improving water quality [55]. The spatial expansion of Makassar City to the Districts of Tamalanrea and Tallo has an impact on increasing the development of large-scale settlements, industries, educational facilities, and trade centers towards the center of economic growth. The accumulation of urban activities has an impact on decreasing environmental quality and increasing water quality pollution in the Tallo watershed. The watershed is mainly located in the central urban area and exhibits the greatest population aggregation [56]. Furthermore, this has an impact on the amount and distribution of rainfall and spatial and temporal falls in watersheds and their runoff times [57].

Field facts found show that the main causes of environmental pollution along the Tallo watershed, namely household waste, garbage, land reclamation and urban activity waste disposal. [58], mention that the anthropogenic activities and natural factors that affect river water quality and can assist in the design of efficient strategies for controlling river water pollution at the watershed scale. Furthermore, productive harmony "between humans and nature depends on the integrity of the ecosystem in terms of their structure, function and capacity to produce goods and services that humans need, including clean and abundant water [59].

The control of environmental pollution and control of spatial use in the Tallo watershed is very important to be implemented through the support of government policies. Watershed management is a growing practice that involves managing land, water, biota and other resources in areas designated for ecological, social and economic purposes [60]. Furthermore, the activities of wastewater facilities located in residential areas due to increasing population, and other socioeconomic variables negatively impact water quality in the watershed [61]. Field facts found show that environmental degradation in the Tallo watershed has a direct relationship with the existence of slums, community poverty, and community behavior. That inequality of space reproduction control, lack of access to economic, sociocultural resources, subsistence economic conditions and inadequate support of infrastructure services has caused slum areas to develop along the Tallo watershed in Makassar City [62]. This fact after being confirmed in the field shows that disparity in infrastructure services and increased economic productivity and weak control of spatial use due to large-scale settlement development, socio-economic activities and slum development are causing environmental degradation and pollution of water quality in the Tallo watershed. The results of the regression analysis of the factors causing a decrease in the environmental quality of the Tallo watershed in the following.

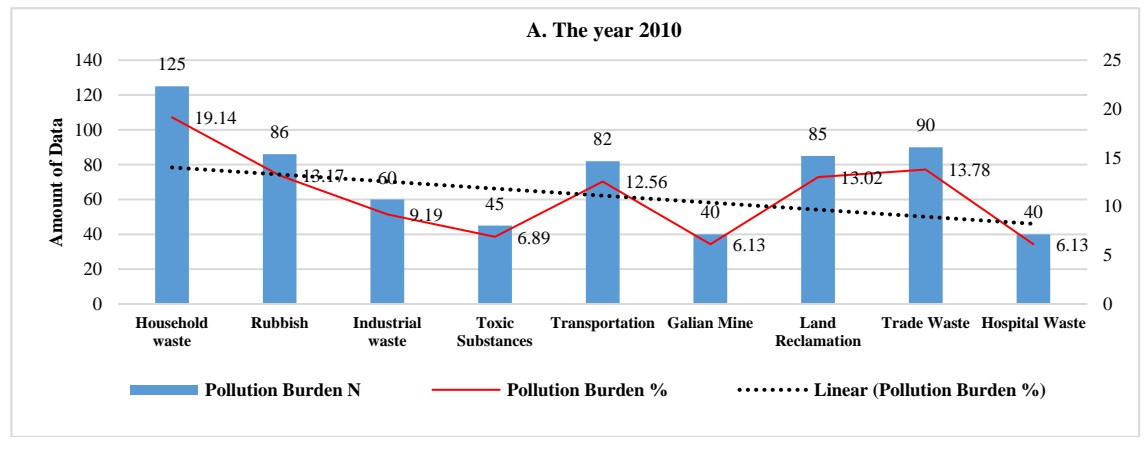

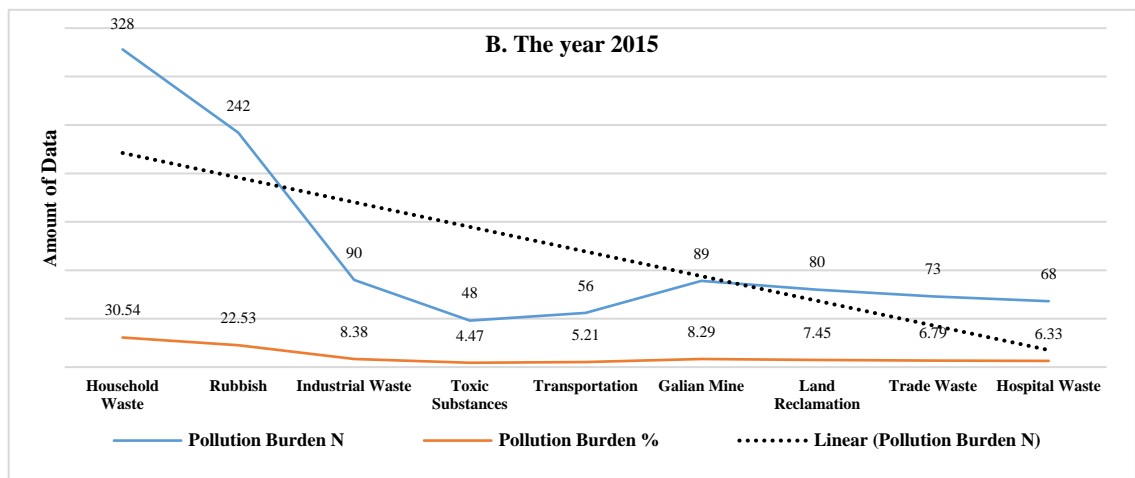

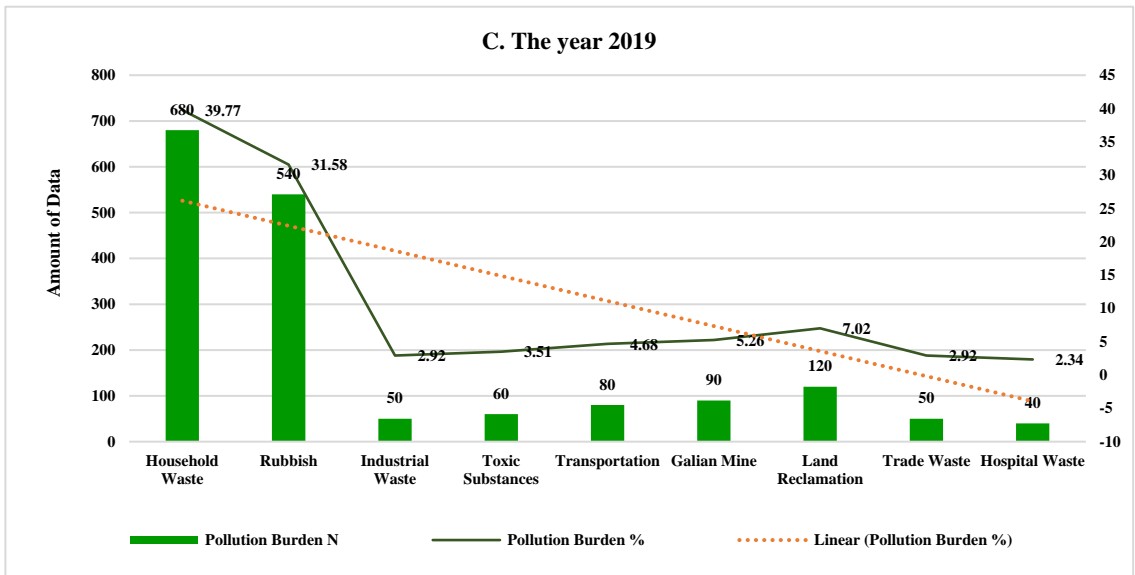

**Figure 9.** Load and Sources of Environmental Pollution in the Tallo River Basin in Makassar City (2010–2019). Source: author elaboration and Data Primer.

The results of the regression analysis of the tested variables illustrate: (i) the relationship between slums and environmental degradation of watersheds by 0.39. The coefficient of determination of the effect of slums is 15.2%, (ii) the relationship of community poverty with a decrease in the environmental quality of watersheds by 0.38. The coefficient of determination of the effect of community poverty by

14.3%, (iii) the relationship of community behavior with a decrease in the environmental quality of the watershed by 0.36. The coefficient of determination of the effect of community behavior by 13.2%. Thus, it can be concluded that the development of slums, poverty and community behavior together have a significant effect on the decline in the environmental quality of the Tallo river basin in Makassar City, with a coefficient of determination/influence of 32.2%.

Strategic steps that require implementation in the field related to the results of the analysis (see Table 2) include: (i) the formulation of programs based on community capacity building, (ii) developing an environmentally conscious culture based on community participation, (iii) setting limits on zonation-based watershed utilization, and (iv) mangrove forest rehabilitation and rejuvenation of slum settlements based on community empowerment. The four strategic steps will require system integration in the formulation of government policies and comprehensive planning mechanisms that favor the poor as residents of slums located in watersheds. Policies and programs aimed at improving the wellbeing of slum dwellers should address comprehensively the underlying structural, economic, behavioral, and service-oriented barriers to good health and productive lives among slum residents [63]. Furthermore, the lack of a law enforcement framework will reduce the effectiveness of policy implementation [64].

**Table 2.** Summary of Results of Associative Hypothesis Testing.

| Correlated Variables | t Count | r Table | Information | $r^2$ |
| --- | --- | --- | --- | --- |
| Slum Settlements Towards Degradation of Environmental Quality in Watersheds ($ryx_1$) | 0.39 | 0.297 | Signifikan | 0.152 |
| Community Poverty Towards a Declining Quality of the Watershed Environment ($ryx_2$) | 0.38 | 0.297 | Signifikan | 0.143 |
| Community Behavior Towards Declining Quality of Watershed Environment ($ryx_3$) | 0.36 | 0.297 | Signifikan | 0.132 |
| Slum Settlement, Poverty and Community Behavior Towards a Declining Quality of the Watershed Environment (R) | 0.566 | f = 3.22 | Signifikan | 0.320 |

*4.4. Management Control and Actions*

System integration in natural resource management is very important to be implemented for the needs of management, pollution control, and anticipation of environmental quality deterioration along the Tallo River Basin in the City of Makassar in the future. Watershed management is considered the most appropriate approach to ensure conservation and sustainability of all land-based resources and to improve living conditions people in the highlands and lowlands [65]. Furthermore, mangrove forest conservation that will be implemented is intended to maintain environmental balance and maintain its function as a physical and biological protector. In its implementation, it is integrated with the development of ecotourism, strengthening community institutional capacity towards increasing community economic productivity. Community members' support for tourism has a crucial function in bridging the link from community empowerment to sustain tourism in a local area, offering the reminder that human society ultimately depends on their natural environment [66].

The implementation of a tourism-based natural resource conservation program is basically intended to encourage increased economic productivity of the poor who are in slum areas along the Tallo river basin (see Figure 10). Five objectives to be achieved in implementing mangrove forest conservation in the Tallo watershed of Makassar City include: (i) protect the natural ecosystem to maintain the balance of the ecosystem, (ii) protect the flora and fauna that is in it, to keep the species from becoming extinct, (iii) protect the ecosystem from damage, whether caused by human factors or natural factors, (iv) protecting the environment to be maintained, and (v) protecting the natural ecosystems of the Tallo watershed. The affirmation of the concept of ecotourism-based conservation developed in the research area is aimed at encouraging the improvement of the welfare of communities located in slums, in the sense that community involvement will have a positive impact on productive

economic efforts and the opening of new jobs. Biodiversity conservation, as an environmental goal, is increasingly recognized to be connected to the socio-economic well-being of the local community [67]. Furthermore, increasing local involvement and participation will help ensure that communities are empowered, and the preservation of natural resources takes place [68].

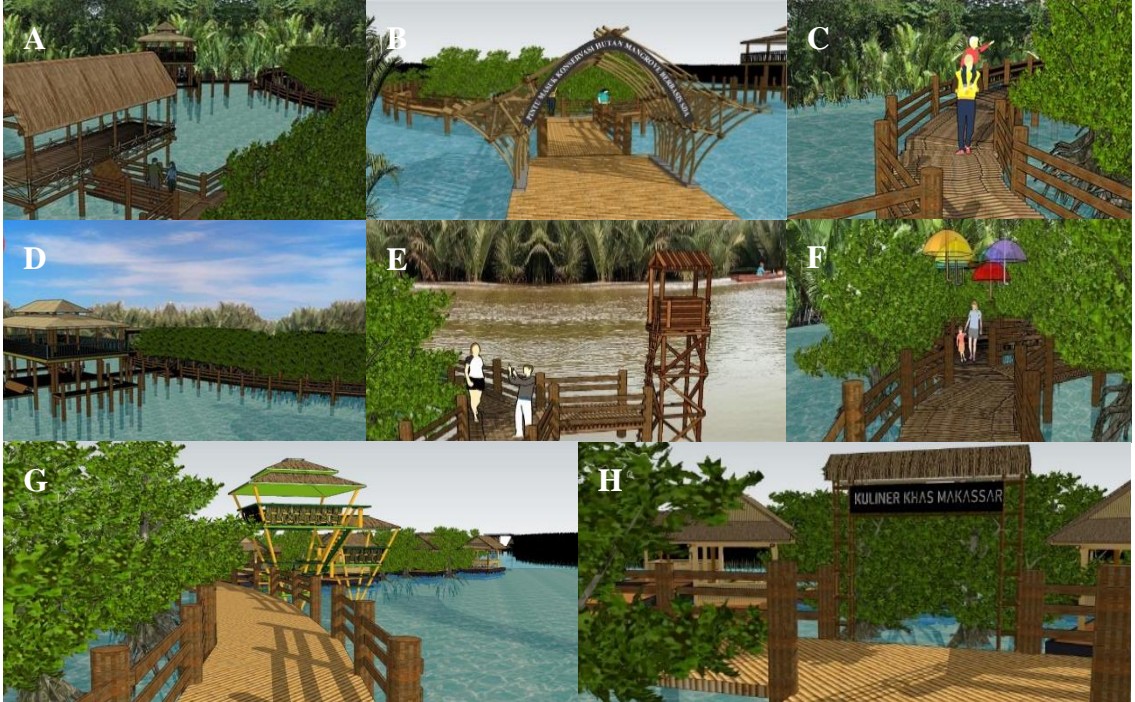

**Figure 10.** Example, Conservation of Tourism-Based Natural Resources and Improvement of Community Economic Productivity in the Tallo Watershed of Makassar City.

In order to support the improvement of economic business activities, in relation to community economic empowerment, the program of activities developed and integrated with the rehabilitation of mangrove forests includes: (a). The construction of the gate (Figure 10B), is intended as a central marker of economic activity activities, as well as an entrance to the tourist area. (b). A culinary economic business that serves traditional food from the local community (Figure 10A,D,H). This economic effort was developed to provide economic opportunities for local communities whose development was facilitated by the government. This effort was developed for the purpose of providing economic benefits for the local community to improve welfare and reduce the unemployment rate of the productive population. (c). A mangrove walkway (Figure 10C,E–G) was developed to provide an atmosphere for visitors to the tour, in addition to enjoying the beauty of nature it also provides an educational process for the community to continue to preserve mangrove forests. Along the walkway, photo spot areas, monitoring towers, direction markers, and some important information related to the preservation of the Tallo watershed are developed.

The program implementation is carried out in several stages, namely: (i) socialization and community preparation, aimed at getting the same perspective among stakeholders, in this case the involvement of each actor's role, (ii) strengthening the capacity of the poor who live in slums, through facilitation and facilitation processes, (iii) training economic efforts based on the potential of local resources oriented to strengthening local economies based on sustainable ecotourism, and (iv) implementation of conservation based on the participation of local communities oriented towards behavioral change and community awareness towards optimizing the utilization of the natural resource potential of watersheds. The four stages of the process will lead to the integration of systems for handling and controlling environmental pollution and saving mangrove forests in a sustainable manner.

Mangrove provides many ecosystem services and has an important role in both the number and the type of ecosystem services [69].

Integration of mangrove forest conservation systems implemented in the study area is focused on increasing the productivity of economic businesses, in the framework of encouraging community welfare and independence improvement (see Figure 11). The concept of system integration in the context of handling the Tallo watershed is realized through the role of the actors, in this case the government in terms of decision making and supervision, private in this case the allocation of funding in the form of corporate social responsibility (CSR) on social and environmental conditions of the community, as well as participation the community in maintaining infrastructure and improving the quality of the environment. Roles and responsibilities of the government include working to improve welfare through increased income. One way to do that is through empowerment that summarizes existing social values in society [70]. The effort was created to optimize the utilization of potential watersheds towards the development of community-based ecotourism in the interests of overcoming the problem of poverty and public awareness towards an environmentally conscious culture. Community-based tourism has been pushed as one of the strategies for poverty alleviation and it might enhance the sustainability of marginalized regions and communities [71]. Furthermore, policies towards redistribution strengthen the skills, resources, and conditions of micro-entrepreneurship, community-based and family, along with a stronger orientation to the domestic market [72].

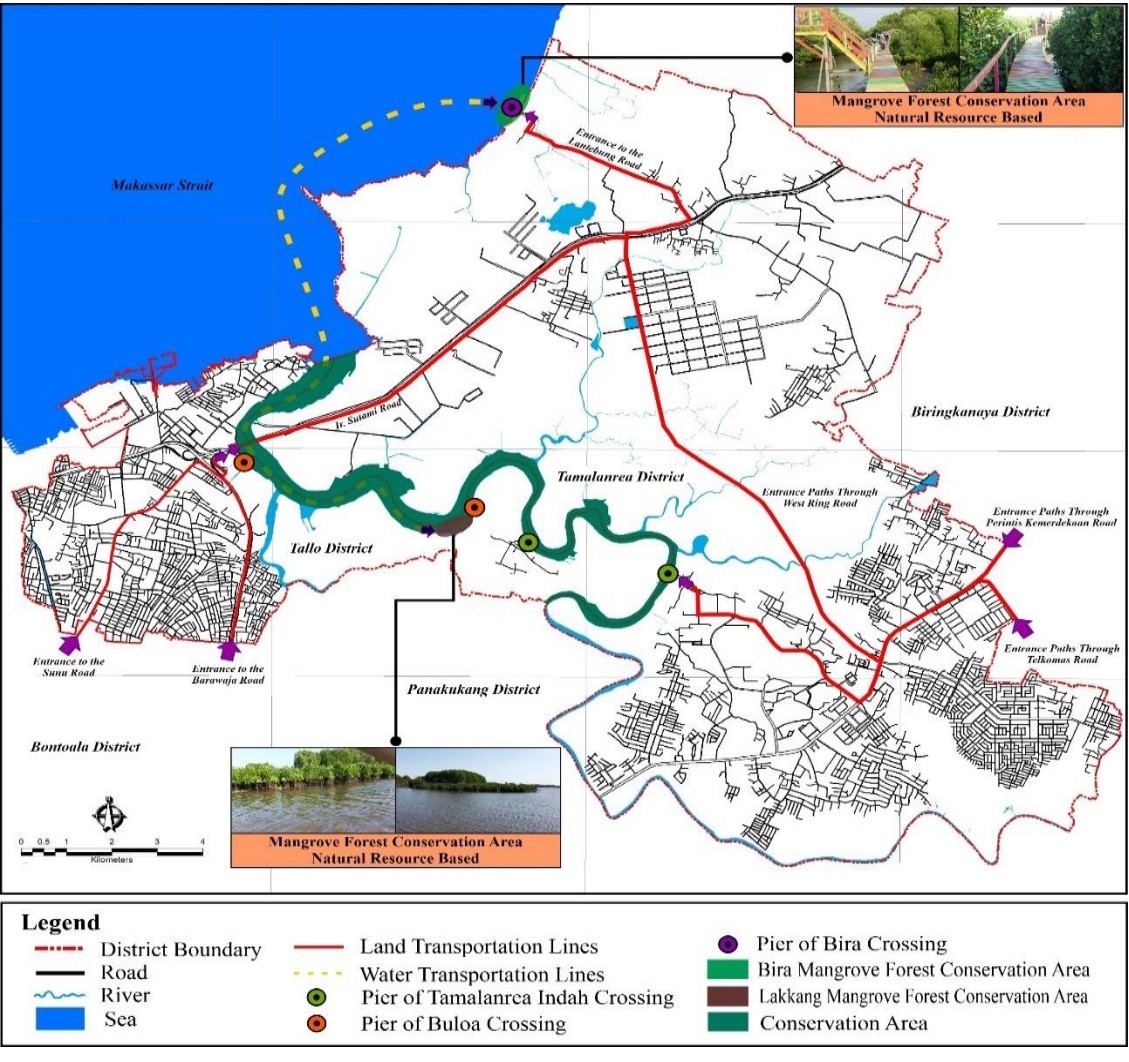

**Figure 11.** Integration of the Natural Resource Conservation System in the Tallo River Basin in Makassar City.

In order to increase the productivity of the community's economic ventures, the components of tourism activities that are developed, are complemented by economic business activities, namely; (i) culinary business, which is managed by the community individually and in groups (ii) preparation of entertainment facilities and tourist facilities, which will absorb local workforce of 564 people, in order to reduce the unemployment rate of productive age, (iii) accommodation facilities will absorb labor of 167 inhabitants and (iv) a mangrove boardwalk with a six-meter wide body path for the purpose of facilitating tourist visitors to enjoy the atmosphere of the mangrove forest. Thus, the integration of a tourism-based mangrove forest conservation system that involves the role of community participation will encourage increased economic effort and community independence to achieve sustainable development goals in the Tallo watershed of Makassar City. Local level participation is very important to achieve global goals, i.e., sustainable development [73].

The integration of the system developed towards the sustainability of watershed conservation and handling of slums is contextualized in human relations with the natural environment by utilizing local wisdom and social capital that has been built up in the community environment. Environmental sociological theories can clarify the connections and interactions between institutional dynamics and their ecological outcomes [74]. Furthermore, local communities adapt sustainable development to their individual contexts and the accumulation of social capital and sustainability will be achieved through the governance and networks available in the community [75]. The path analysis results are shown in the following.

Conservation of natural resources based on economic empowerment will have an impact on increasing community economic productivity and restoring environmental quality towards the sustainability of the Tallo watershed ecosystem (see Figure 12). The results of the path analysis illustrate the relationship between conservation of natural resources, economic empowerment, community capacity building, the productivity of economic enterprises and ecosystem sustainability, and affirmation; (i) the relationship or correlation between natural resource management and economic empowerment by 0.802, (ii) the relationship or correlation between natural resource management and community capacity strengthening by 0.602, and (iii) the relationship or correlation between economic empowerment and community capacity strengthening by 0.542.

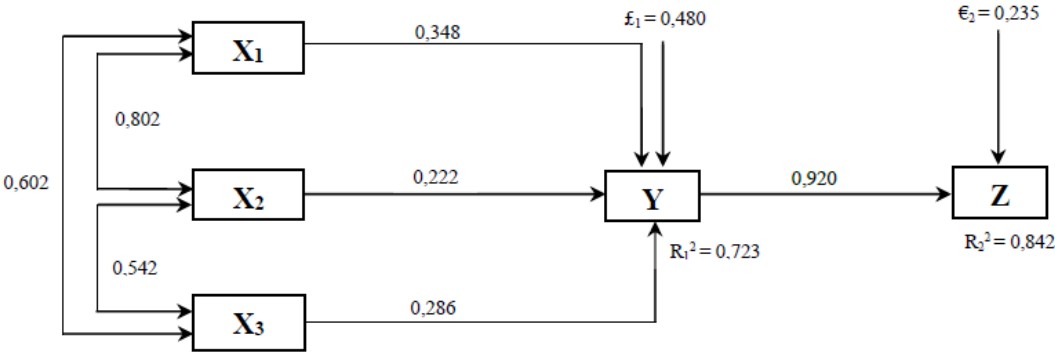

**Figure 12.** The Relationship of Natural Resource Management, Economic Empowerment, Strengthening Community Capacity, Towards Economic Business Productivity and Ecosystem Sustainability.

The direct effect of natural resource conservation on increasing the productivity of community economic ventures by 0.121% or 12.1%. The direct effect of economic empowerment on the productivity of economic enterprises was 0.049% or 4.93%. The direct effect of strengthening community capacity on increasing the productivity of economic ventures was 0.0812% or 8.18%. The results confirm that the conservation of natural resources, economic empowerment, and community capacity building are positively correlated to increasing the productivity of economic enterprises and the sustainability of watershed ecosystems. Ecosystem economics and biodiversity will helped pave the way for the "green economy" to emerge as the dominant policy approach to achieve environmental protection

and economic growth [76]. Furthermore, policymakers must maximize the relationship between the social economy, the capacity of society and other sectors, and policies that encourage entrepreneurship through the role of the private sector as an integrated whole system [77].

The indirect effect of natural resource management through strengthening community capacity on economic business productivity by 0.047% or 4.70%. The indirect effect of economic empowerment through the conservation of natural resources on the productivity of economic enterprises is 0.0619 or 6.19 percent. The indirect effect of natural resource management through economic empowerment on increasing the productivity of economic ventures by 0.054% or 5.40%. The indirect effect of economic empowerment through natural resource management is 0.059% or 5.90%. The indirect effect of natural resource management is through strengthening community capacity by 0.034% or 3.40%. The indirect effect of strengthening community capacity through economic empowerment is 0.00215% or 0.22%.

The total effect was 44.74%. The remaining influence or residue (the effect of other variables on the productivity of economic enterprises not examined is 0.5526% or 55.26%. Furthermore, the direct effect of economic business productivity on the sustainability of watershed ecosystems is 0.8464%. The remaining influence or residue (the influence of other variables towards ecosystem sustainability that is not examined) by 0.1536% or 15.36%, these results indicate that there is a strengthening of Y by 44.74% and the effect of Y on Z by 55.26%. Thus, it can be concluded that natural resource management through watershed conservation that is integrated with economic empowerment has a positive contribution to the sustainability of the environmental ecosystem. The results obtained support the triple bottom-line approach namely, integrating economic, environmental and social factors [78]. Furthermore, the sustainable circular economy not only adopts an environmental perspective, but also considers economic and social performance [79]. The results of the path analysis, confirming that there is a need for a stable and sustainable community economic circulation through optimizing the utilization of natural resources and empowering the community's economy, are the key to managing slums in the direction of creating inclusive and sustainable community harmony in order to reduce socio-economic disparities.

### 4.5. Community Based Economic Empowerment in the Management of Slums

Economic empowerment has significance for the urban poor in slums located in watersheds. These communities need to move out of poverty and destitution through coaching and mentoring by various experts, and at the same time they would assure the ecosystem functioning of urban rivers [80]. The focus of community economic empowerment in the study area is oriented to strengthening community capacity, the ability to manage business risks, being able to adapt to changes and seize opportunities to develop economic businesses in a sustainable manner. Increased awareness that quality of life with communities and institutions can do better to increase community capacity, manage risk, accept change, and seize opportunities [81]. A scheme of community economic empowerment based on the handling of slums is shown in the following.

The direction of community economic sustainability based on natural resources carried out through watershed conservation in relation to the handling of slums (see Figure 13), implemented in several stages, includes: First, the conservation of the Tallo watershed is oriented to efforts to optimize the utilization of natural resource potentials and environmental preservation by considering the socio-economic conditions of the community through strengthening institutional capacity, community participation and the use of social knowledge based on local wisdom. Green economic development is influenced by local wisdom, economic potential, and role community institutions [82]. Second, economic empowerment is carried out through a process of facilitation, assistance, and mapping of economic potential that will be achieved by the community in the direction of ecotourism-based productive economic business training which is carried out by the community independently. Empowering indigenous peoples to embrace ecotourism control has been advocated as an integral component of sustainable tourism [83].

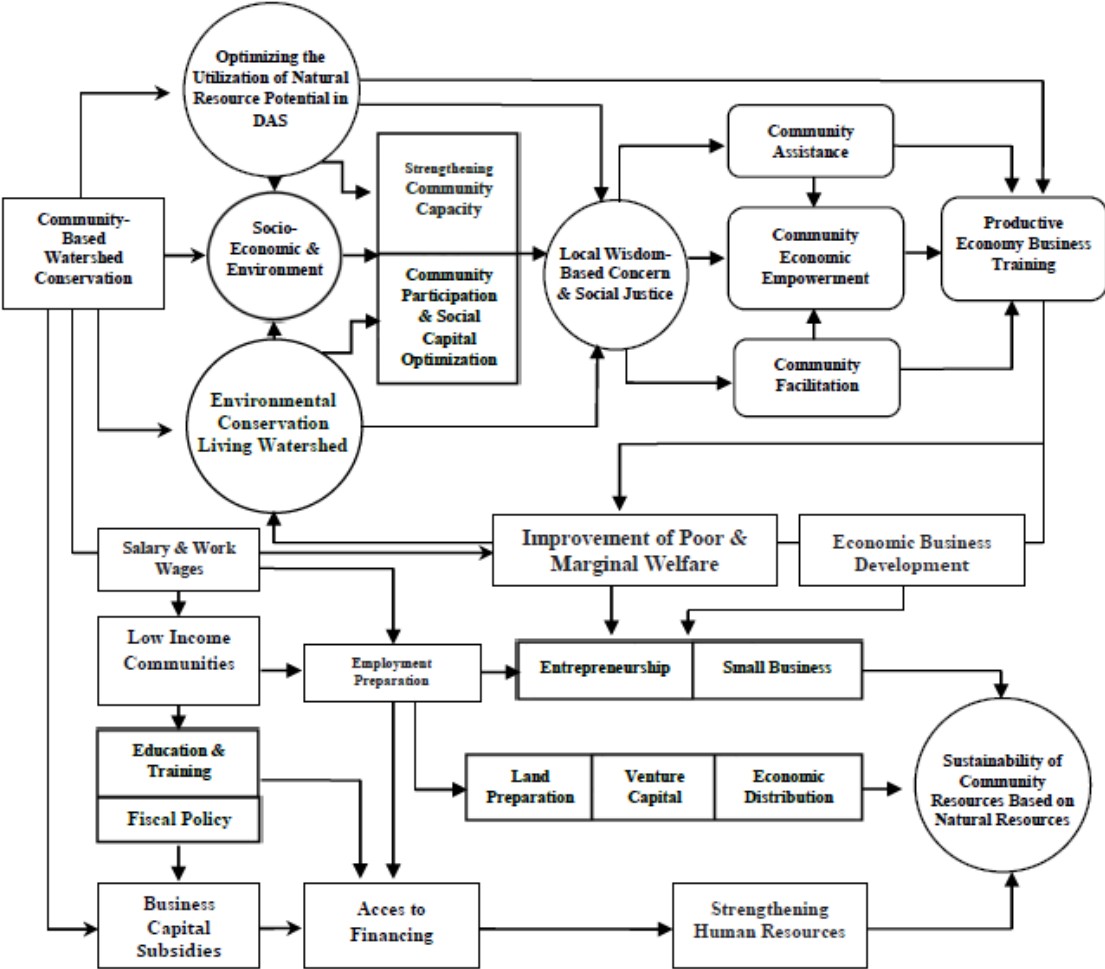

**Figure 13.** Community Economic Empowerment Based on Management of Settlements and Watershed Conservation.

Third, the development of economic businesses is carried out by considering; utilization of local labor, entrepreneurship training, business development, land preparation, business capital assistance and fairness in economic distribution. Policies that lead to effective training for entrepreneurs, by providing incentive support and opportunities to start a business [84]. Furthermore, a person determined to become an entrepreneur must evaluate not only the external, but also personal circumstances [85]. For people who have entrepreneurial talent are facilitated through training, guidance and support for venture capital assistance. Whereas those who do not have business talent are used as labor in the productive economic enterprise that develops.

Fourth, government policy support, realized in the form of education and training, fiscal policy, business capital support, access to funding and strengthening human resources. The main factors affecting economic resilience are the availability of social communities and formal economic institutions, institutional capacity, availability of production centers and social capacity [86,87]. Furthermore, this policy will encourage public confidence in the government by applying the principles of social justice, equitable development, and reducing the gap to stability in development [88]. This result confirms that economic empowerment carried out through system integration and the collaboration of actors, namely the government, private sector, and the community, will encourage the sustainability of economic enterprises towards increasing the productivity of community businesses and improving the quality of the environment of slums as a tangible manifestation of the successful implementation of a watershed resource conservation program community based river.

## 5. Conclusions

The development of slums, poverty, and community behavior significantly influence the environmental quality of the watershed. Natural resource management through watershed conservation and management of slums will be successful in its implementation if followed by the role of active community participation. Watershed conservation based on economic empowerment integrated with ecotourism development through involving the role of community participation will encourage increased economic effort and community independence towards the achievement of sustainable development goals in Makassar City. Furthermore, the implementation of strategic policies and the integration of the Tallo watershed management system through the optimization of the utilization of natural resources and the empowerment of the community's economy is the key to handling slum settlements towards a stable and sustainable economic circulation of the community.

The economic sustainability of the community based on natural resources in the handling of slums in watersheds is oriented towards economic empowerment based on system integration and the collaboration of actors, including the government, the private sector, and the community. Economic empowerment is carried out through the facilitation and mapping of economic potential that will be achieved by the community in the direction of ecotourism-based productive economic enterprise training, which is carried out independently by the community. Government policy support, realized in the form of education and training, fiscal policy, business capital support, access to funding and strengthening human resources. System integration and the collaboration of actors, namely the government, private sector, and the community will drive the sustainability of economic enterprises towards increasing the productivity of community businesses and improving the quality of the environment of slums and tangible manifestations of the successful implementation of community-based watershed conservation programs in the direction of creating integrated development, social cohesion, environmental quality improvement, and economic sustainability of the community.

**Author Contributions:** B.S. and S.Y. compile research; B.S. and H.H.S. designed the methodology; H.S. processes data; H.H.S. and S.Y. contribute materials/methods/analysis tools; B.S. and H.S. analyze data; H.S. contributes to data checking; B.S., S.Y., H.H.S. and H.S. wrote and revised the draft. All authors have read and agreed to the published version of the manuscript.

**Funding:** This research was funded by the Government of the Republic of Indonesia through the Ministry of Research and Technology in the form of development research grant assistance.

**Acknowledgments:** We are grateful for the participation of stakeholders in contributing ideas in carrying out this study. Thank you to the Ministry of Research and Technology of the Republic of Indonesia for their support and financial assistance in carrying out this research.

**Conflicts of Interest:** The authors declare no conflict of interest.

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
