# Peer review of "Natural Resource Conservation Based on Community Economic Empowerment: Perspectives on Watershed Management and Slum Settlements in Makassar City, South Sulawesi, Indonesia"

_land, doi:10.3390/land9040104_

Round 1
Reviewer 1 Report
As urbanization going on, understanding of effects of it on environment and natural resources caused by increased population, especially for people formed slums, is very important for both land use and landscape. In this case, the topic of this manuscript is interesting and falls into the scope of Land. However, there are many problems in the current version, and I suggest that authors should revise it intensively and then resubmit. Also, English must be improved.
Specific comments
Introduction
This section you should reintegrate. There are many repeated sentences. Such as L 40-42 and L67-70, L51-52, L56-57, L 61-62, and L64-65.
The logical of writing this section as I think should be as following
Why there is a urbanization around the world?
How did the world achieve urbanization and what urbanization status of the world was?
What effects of urbanization have? Including effects of urbanization on scarcity of resources, regional disparity, metropolitan cities, slums and livelihoods and environment, etc..
Beginning with your country urbanization, you should describe similarities and dissimilarities with the world’s or other countries’ urbanization, and then put forward the specific issues related to your study and the reasons for your study.
Method
As I think, Line 186-223 and section 3.1 should regarded as method of slums selection
Section 2.1, the samples selected are not introduced clearly. You should give the specific social methods related to samples. The number of samples?
Results
Section 3 has many repeated sentences, and L257-288 were not listed in right place due to lack of logicality.
Results section should be as following:
Section 3.2 should be section 3.1
Section 3.2 should be causes of environmental pollution
Section 3.3 should be controlling or management measures
Author Response
Dear Revewer.
- Improvements to the instructions we have done as requested. (result of improvement on page 1-5).
- The structure of the discussion on the methodology, we have refined according to the input provided, specifically related to the elaboration of research samples, how to take samples and so on (pages 7-16).
- We have also correted the structure in the study result as reguested (page 16-30).
Thanks you again for your input for improvements, our article.
Author

Reviewer 2 Report
Comments and Suggestions for Authors:
- Title should be completed like “Natural Resource Conservation Based on Community Economic Empowerment: Perspectives on Watershed Management and Slum Settlements in Makassar City of Indonesia”.
- Abstract should be revised based on research findings not focused on the literature.
- ‘Introduction’ did not reflect the study objectives so it should be rearranging in light of the focus of study components.
- Missing the sequence of literature in the introduction section.
- Use recent published documents/articles in case of data/statistics-oriented information like references 5, 11, 12, 17, 18, 21, 25, 26, etc.
- Avoid the use of British and American spellings. Must be used a common English grammar and spellings, not both.
- Moderate English changes required. Minor spell check required. Minor editing of English language and style required.
- Use migration not the population mobility.
- Missing the information of why choose Indonesia as a study country. Required a brief description of the economy, environment, demographics, urbanization and resource conservation patterns.
- Always use past tense in methodology and results sections.
- For the “Conceptual framework” should require a separate paragraph under the same heading.
- There is lack of research questions.
- Write ‘ha’ not ‘Ha’ for hectares.
- “Materials and Methods” section should be rewritten and rearrange. As for example: 2.1: Profile of the study area; 2.2: Data collection methods (2.2.1: primary data collection, 2.2.2: selection of the respondents, 2.2.3: survey (questionnaire, KII, FGD, etc.), 2.2.4: secondary data/literature review); 2.3: Maps preparation; 2.4: Data analysis including brief of all equations used in the paper with references of sources of the equations.
- “Results and Discussion”: Always use the past tense to present the results.
- Figure 3 & 4 should be under a heading.
- Use ‘USD’ instead of ‘Rp’ for monetary values.
- Results should be shortened but brief and sub-divided into small headings.
- Required a brief description of Figures 5, 6, 7, 8 and 9 (description of each picture with a labelling of a, b, c, d,…etc.).
- Conclusion: missing the integration of the results into policy context for suggestions or recommendations.
- References: recheck the format of referencing.
Author Response
Dear Reviewer:
Thanks you for your input, We have made same improvements, as follows:
- We have adjusted the title as reguested
- We have fixed the abstract according to the result to the research reguired.
- We have fixed the suggestions for improvement in the instroduction (pages 1-5).
- The structure of our research methodology has been improved as suggested (pages 7-16).
- We have improved the study result and discussion (pages 16-30).
- Our Conclusion have been improved upon reguest.
- Our references have been improved upon reguest.
Thanks you for the advice and input.
Author

Reviewer 3 Report
Good paper which tackles natural resources conservation in a very challenging environment - slum settlements.
Please include a section on Sustainable Development Goals and slum settlements.
In the conclusion please advocate better your proposed model based on empowerment, integrated urban development, and social cohesion, as well as the environmental upgrading and economic development so eagerly desired by the people.
Author Response
Dear Reviewer.
Thanks you for your input.
- We have fixed the abstract as regusted.
- We have corected the conclusion in accordance with the suggestions and input.
Thanks you for your infut.
Author

Round 2
Reviewer 2 Report
Comments to the authors: see attached file

Author Response
Dear Reviewer
As per your suggestions and input, we have made improvements to our artickle as, Follows:
- Our abstract improvement have been improved according to suggestions and input.
- Our Introduction has been reduced according to suggestions and input (pages 1-3).
- We have reduced the explanation of the profile of the study area (pages 5-7).
- Our Methods section has been improved according to suggestions and input (pages 8-10).
- Explanation of Figure 6 (pages 14-15), Figure 8 (pages 18-19), and improvements to Figure 10 (pages 23).
- We have adjusted the reference again
With this we extend our highest appreciation for the suggestions and input
Author.
